

# Evaluating and Improving the Reliability of Gas-Phase Sensor System Calibrations Across New Locations for Ambient Measurements and Personal Exposure Monitoring

Sharad Vikram [1], Ashley Collier-Oxandale [2], Michael Ostertag [1], Massimiliano Menarini [1], Camron Chermak [1], Sanjoy Dasgupta [1], Tajana Rosing [1], Michael Hannigan [2], and William G. Griswold [1]

[1]Department of Computer Science & Engineering, University of California, San Diego
[2]Environmental Engineering Program, University of Colorado, Boulder

**Correspondence:** William B. Griswold (wgg@cs.ucsd.edu)

**Abstract.**

Advances in ambient environmental monitoring technologies are enabling concerned communities and citizens to collect data to better understand their local environment and potential exposures. These mobile, low-cost tools make it possible to collect data with increased temporal and spatial resolution providing data on a large scale with unprecedented levels of detail. This type of data has the potential to empower people to make personal decisions about their exposure and support the development of local strategies for reducing pollution and improving health outcomes.

However, calibration of these low-cost instruments has been a challenge. Often, a sensor package is calibrated via field calibration. This involves colocating the sensor package with a high-quality reference instrument for an extended period, and then applying machine learning or other model fitting technique such as multiple-linear regression to develop a calibration model for converting raw sensor signals to pollutant concentrations. Although this method helps to correct for the effects of ambient conditions (e.g., temperature) and cross-sensitivities with non-target pollutants, there is a growing body of evidence that calibration models can overfit to a given location or set of environmental conditions on account of the incidental correlation between pollutant levels and environmental conditions, including diurnal cycles. As a result, a sensor package trained at a field site may provide less reliable data when moved, or *transferred*, to a different location. This is a potential concern for applications seeking to perform monitoring away from regulatory monitoring sites, such as personal mobile monitoring or high-resolution monitoring of a neighborhood.

We performed experiments confirming that transferability is indeed a problem and show that it can be improved by collecting data from multiple regulatory sites and building a calibration model that leverages data from a more diverse dataset. We deployed three sensor packages to each of three sites with reference monitors (nine packages total), and then rotated the sensor packages through the sites over time. Two sites were in San Diego, CA, with a third outside of Bakersfield, CA, offering varying environmental conditions, general air quality composition, and pollutant concentrations. When compared to prior single-site calibration, the multi-site approach exhibits better model transferability for a range of modeling approaches. Our experiments also reveal that random forest is especially prone to overfitting, and confirms prior results that transfer is a significant source of



both bias and standard error. Bias dominated in our experiments, suggesting that transferability might be easily increased by detecting and correcting for bias.

Also, given that many monitoring applications involve the deployment of many sensor packages based on the same sensing technology, there is an opportunity to leverage the availability of multiple sensors at multiple sites during calibration. We

contribute a new neural network architecture model termed *split-NN* that splits the model into two-stages, in which the first stage corrects for sensor-to-sensor variation and the second stage uses the combined data of all the sensors to build a model for a single sensor package. The split-NN modeling approach outperforms multiple linear regression, traditional 2- and 4-layer neural network, and random forest models.

## 1 Introduction

As the use of low-cost sensor systems for citizen science and community-based research expands, improving the robustness of calibration for low-cost sensors will support these efforts by ensuring more reliable data and enabling a more effective use of the often-limited resources of these groups. These next-generation technologies have the potential to reduce the cost of air quality monitoring instruments by orders of magnitude, enabling the collection of data at higher spatial and temporal resolution, providing new options for both personal exposure monitoring and communities concerned about their air quality. (Snyder et al.,

2013). High resolution data collection is important because air quality can vary on small temporal and spatial scales (Monn et al., 1997; Wheeler et al., 2008). This variability can make it difficult to estimate exposure or understand the impact of local sources using data from existing monitoring networks (Wilson et al., 2005), which provide information at a more regional scale. Furthermore, studies have highlighted instances where air quality guidelines have been exceeded on small spatial scales, in so called 'hot spots' (Wu et al., 2012). This may be of particular concern for environmental justice communities, where residents

are unknowingly exposed to higher concentrations of pollutants due to a lack of proximity to local monitoring stations. One group using low-cost sensors to provide more detailed and locally specific air quality information is the Imperial County Community Air Monitoring Network (English et al., 2017). The goal of this network of particulate monitors is to help inform local action (e.g., keeping kids with asthma inside), or open the door to conversations with regulators (English et al., 2017). In another example, researchers are investigating the potential for wearable monitors to improve personal exposure estimates

(Jerrett et al., 2017).

The increasing use of low-cost sensors is driving a growing concern regarding data quality (Clements et al., 2017). Low-cost sensors, particularly those designed to detect gas-phase pollutants, are often cross-sensitive to changing environmental conditions (e.g., temperature, humidity, and barometric pressure) and other pollutant species. Much work has gone into exploring calibration methods, models, and techniques that incorporate corrections for these cross-sensitives to make accurate

measurements in complex ambient environments (Spinelle et al., 2014, 2015b, 2017; Cross et al., 2017; Sadighi et al., 2018; Zimmerman et al., 2018). While the methods of building (or *training*) calibration models differ, these studies have all utilized colocations with high-quality reference instruments in the field, instruments such as Federal Reference Method or Federal Equivalent Method monitors (FRM/FEM) (Spinelle et al., 2014, 2015b, 2017; Cross et al., 2017; Sadighi et al., 2018; Zim-



merman et al., 2018). This colocated data allows accurate calibration models to be built for the conditions that the sensors will experience in the field (e.g., diurnal environmental trends and background pollutants). A recurring observation has been that laboratory calibrations, while valuable for characterizing a sensor's abilities, perform poorly compared to field calibrations likely due to an inability to replicate complex conditions in a chamber (Piedrahita et al., 2014; Castell et al., 2017).

Recently, researchers have begun to explore calibrating sensors in one location and testing them in another, called *transfer*. Often, a decrease in performance is seen in new locations where conditions are likely to differ from the conditions of calibration. In one study, researchers testing a field calibration for electrochemical $SO_2$ sensors from one location in Hawaii and at another location also in Hawaii found a small drop in correlation between the reference and converted sensor data (Hagan et al., 2018). This was attributed to the testing location being a generally less polluted environment (Hagan et al., 2018). In a study

that involved calibration techniques for low-cost metal-oxide $O_3$ sensors and non-dispersive infrared $CO_2$ sensors in different environments (e.g., typical urban vs. a rural area impacted by oil and gas activity), researchers found that simpler calibration models (i.e., linear models), although generally lower in accuracy, performed more consistently (i.e., transferred better) when faced with significant extrapolations in time or typical pollutant levels and sources(Casey and Hannigan, 2018). In contrast, more complex models (i.e.,artificial neural networks) only transferred well when there was little extrapolation in time or

pollutant sources. A study utilizing electrochemical CO, NO, $NO_2$, and $O_3$ sensors found that performance varied spatially and temporally according to changing atmospheric composition and meteorological conditions (Castell et al., 2017). This team also found calibration model parameters differed based on where exactly a single sensor node was colocated (i.e., a site on a busy street verses a calm street), supporting the idea that these models are being specialized to the environment where training occurred (Castell et al., 2017). In a recent study targeting this particular issue with low-cost sensors, electrochemical NO and

$NO_2$ sensors were calibrated at a rural site using multivariate linear regression model, support vector regression models, and a random forest regression model. The performance of these models was then examined at two urban sites (one background urban site and one near-traffic urban site). For both sensor types, random forests were found to be the best-performing models, resulting in mean averages errors between 2–4 ppb and relatively useful information in the new locations (Bigi et al., 2018). One important note from the authors is that both sensor signals were included in the models for NO and $NO_2$ respectively, potentially

helping to mitigate cross interference effects (Bigi et al., 2018). In another recent study, researchers also compared several different calibration model types, as well as the use of individualized verses generalized models and how model performance is affected when sensors are deployed to a new location (Malings et al., 2018). An individualized model is a model for a sensor based on its own data, whereas a generalized model combines the data from all the sensors of the same type being calibrated. The researchers found that the best-performing and most robust model types varied by sensor type; for example,

simpler regression models performed best for electrochemical CO sensors, whereas more complicated models, such as artificial neural networks and random forest models, resulted in the best performance for $NO_2$. Despite the varied results, in terms of the best performing model types, the researchers observed that across the different sensor types tested, generalized models resulted in more consistent performance at new sites than individualized models despite having slightly poorer performance during the initial calibration (Malings et al., 2018). If this observation holds across sensor types and the use in other locations, it could

help solve the problem of scaling up sensor networks allowing for much larger deployments.





The mixed results and varying experimental conditions of these studies highlight the need for a more comprehensive understanding of how and why calibration performance degrades when sensors are moved. A better understanding could inform potential strategies to mitigate these effects. As recent research has successfully applied advanced machine learning techniques to improve sensor calibration models (Zimmerman et al., 2018; De Vito et al., 2009; Casey et al., 2018), we believe these techniques could also be leveraged in innovative ways to improve the transferability of calibration models.

This paper contributes an extensive transferability study as well as new techniques for data collection and model construction to improve transferability. We hypothesize that transferability is an important issue for sensors that exhibit cross-sensitivities. Based on the hypothesis that the increased errors under transfer are due to overfitting, we propose that training a calibration model on multiple sites will improve transfer. Finally, we propose that transfer can be further improved with a new modeling method, split-NN, that can use the data from multiple sensor packages trained at multiple sites to train a single sensor's model.

As many previous studies studied colocation with reference measurements in one location and a validation at a second location, we designed a deployment that included triplicates of sensor packages colocated at three different reference monitoring stations and then rotated through the three sites – two near the city of San Diego, CA and one in a rural area outside of Bakersfield, CA. This allows for further isolating the variable of a new deployment location. The analysis focuses on data from electrochemical $O_3$ and $NO_2$ sensors, although other sensor types were deployed and used in the calibration, analogous to (Bigi et al., 2018). These pollutants are often of interest to individuals and communities given the dangers associated with ozone exposure (Brunekreef and Holgate, 2002), and nitrogen dioxide's role in ozone formation. In studying these pollutants, we are adding to the existing literature by examining the transferability issue in relation to electrochemical $O_3$ and $NO_2$ sensors, which are known to exhibit cross-sensitive effects (Spinelle et al., 2015a). We compare the transferability of multiple linear regression models, neural networks, and random forest models. Based on these measurements and comparisons, we contribute the following results:

– Our experiments confirm that calibration model error increases when the model is transferred to a new location. This holds over all the modeling methods we tested.

– Much of the error under transfer was bias error. Given the simple structure of bias error, this suggests that transferability might be easily increased by detecting and correcting for bias.

– Although random forest methods perform extremely well when trained and tested at a single site, its advantages are largely lost under transfer.

– Training a calibration model on multiple sites improves transferability.

– Finally, for applications involving the deployment of many sensor packages based on the same sensing technology, we show it is possible to improve transferability by exploiting the presence of multiple sensors at multiple sites during field calibration. We contribute a new model training method via a two-stage "split" neural network. The first stage corrects for deviations in sensor performance (essentially bias) and the second stage combines all the data of all the sensors to build a model for a single sensor package. This modeling approach outperforms other approaches.



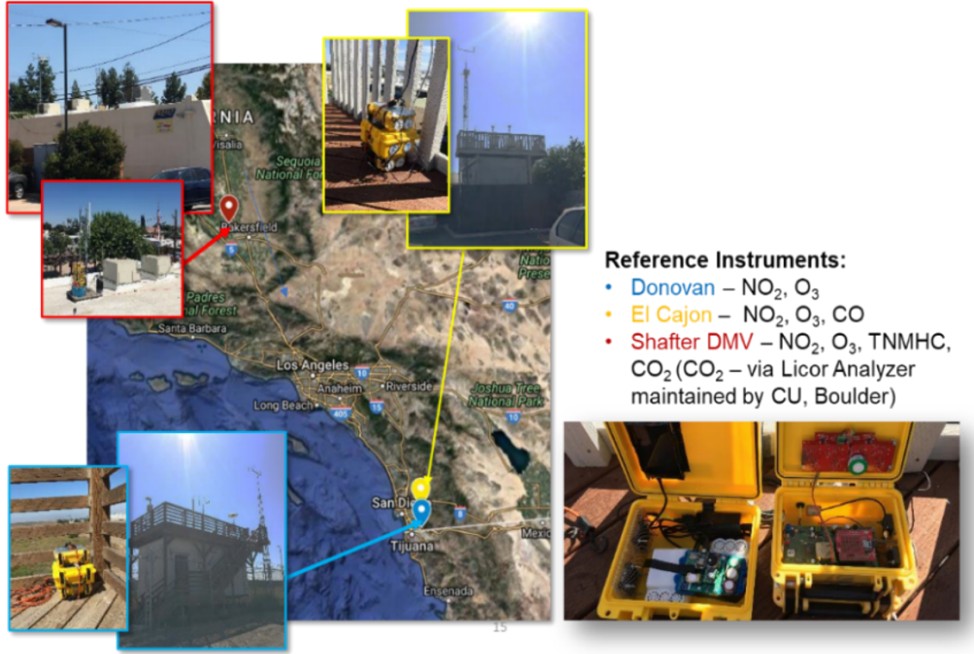

**Figure 1.** Map and images of deployment locations. Shafter DMV (red) was located 250 mi away from Donovan (blue) and El Cajon (yellow), which were located in San Diego, CA. (bottom right) Deployment containers configuration for the extended deployment. Each container has active ventilation to keep the internal conditions equivalent to the ambient environment.

## 2 Methods

### 2.1 Sampling Sites

For this deployment, we coordinated with three regulatory monitoring sites and rotated sensor packages through each site over the course of approximately six months. Each monitoring site included reference instruments for $NO_2$ and $O_3$, among others. The first site was in El Cajon, CA, in a suburban area east of San Diego, CA near an elementary school and a major highway (El Cajon Site). The second site was directly south 15 miles in the south east corner of San Diego, a more rural area approximately two miles from the border crossing for heavy duty vehicles at Otay Mesa (Donovan Site). The third site was in rural Shafter, CA, 250 miles to the north near Bakersfield. It is considerably inland compared to the other sites with nearby agriculture as well as oil and gas extraction activities. We expected to see a unique environmental profile (i.e., temperature, humidity, and barometric pressure) at the Shafter site due to being considerably more inland, where weather would be more dominated by the desert ecosystem rather than the ocean ecosystem. We also expected to see unique emission profiles among the sites. Donovan was expected to show higher truck emissions due to the presence of heavy duty vehicles, potentially idling for long periods of time, while Shafter was expected to be affected by emissions from its nearby oil and gas activity. We expected the El Cajon site's emissions profile to resemble that of a typical urban/suburban site. This variety of environmental and emissions profiles



would allow us to meaningfully test for transferability, in particular to assess to what degree a calibration model trained on one site would overfit for the other sites.

## 2.2 The MetaSense System

### 2.2.1 Hardware Platform

A low-cost air quality sensing platform was developed to interface with commercially available sensors, initially described in Chan et al. (2017). The platform was designed to be mobile, modular, and extensible, enabling end users to configure the platform with sensors suited to their monitoring needs. It interfaces with the Particle Photon or Particle Electron platforms, which contain a 24 MHz ARM Cortex M3 microprocessor and a Wi-Fi or 3G cellular module, respectively. In addition, a Bluetooth Low Energy (BLE) module supports energy efficient communication with smartphones and other hubs with BLE

connectivity. The platform can interface with any sensor that communicates using standard communication protocols (i.e. analog, I2C, SPI, UART) and supports an input voltage of 3.3 V or 5.0 V. The platform can communicate results to nearby devices using BLE or directly to the cloud using Wi-Fi or 2G/3G cellular, depending on requirements. USB is also provided for purposes of debugging, charging, and flashing the firmware. The firmware can also be flashed or configured over the air. An SD card slot provides the option for storing measurements locally, allowing for completely disconnected and low-power

operation.

Our configuration utilized electrochemical sensors for traditional air quality indicators ($NO_2$, CO, $O_3$), nondispersive infrared sensors for $CO_2$, photoionization detectors for volatile organic compounds (VOCs), and a variety of environmental sensors (temperature, humidity, barometric pressure). The electrochemical sensors ($NO_2$: Alphasense $NO_2$-A43F, $O_3$: Alphasense $O_3$-A431, and CO: Alphasense CO-A4) are mounted to a companion analog front end (AFE) from Alphasense, which assists with

voltage regulation and signal amplification. Each sensing element has two electrodes which give analog outputs for the working electrode (WE) and auxiliary electrodes (AE). The difference in signals is approximately linear with respect to the ambient target gas concentration but have dependencies with temperature, humidity, barometric pressure, and cross-sensitivities with other gases. The electrochemical sensors generate an analog output voltage, which is connected to a pair of analog-to-digital converters (ADCs), specifically the TI ADS1115, and converted into a digital representation of the measured voltage, which is

later used as inputs for our machine learning models.

Modern low-cost electrochemical sensors offer a low cost and low power method to measure pollutants, but currently available sensors are more optimized for industrial applications than air pollution monitoring: the overall sensing range is too wide and the noise levels are too high. For example, the AlphaSense A4 sensors for $NO_2$, $O_3$, and CO have a measurement range of 20, 20, and 500 ppm, respectively, which is significantly higher than the unhealthy range proposed by the United States Air

Quality Index. Unhealthy levels for $NO_2$ at 1-hour exposure range from 0.36 – 0.65 ppm, $O_3$ at 1-hour exposure from 0.17 – 0.20 ppm, and CO at 8-hour exposure from 12.5 – 15.4 ppm (Uniform Air Quality Index (AQI) and Daily Reporting, 2015). Along with the high range, the noise levels of the sensors make it difficult to distinguish whether air quality is good. Using the analog front end (AFE) offered by Alphasense, the noise levels for $NO_2$, $O_3$, and CO have standard deviations of 7.5 ppb,



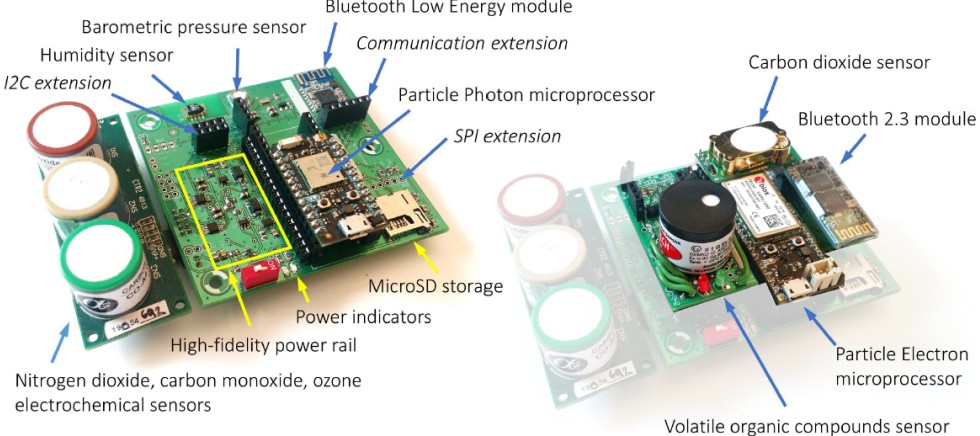

**Figure 2.** Labeled MetaSense Air Quality Sensing Platform. (Left) Modular, extensible platform in standard configuration with $NO_2$, $O_3$, and CO electrochemical sensors. (Right) Additional modules that can be added to the board for additional measurement capabilities.

7.5 ppb, and 10 ppb, respectively. These standard deviations are large compared to observed signal levels for $NO_2$ and $O_3$ measurements, which ranged between 0 – 35 ppb and 12 – 60 ppb, respectively, during the 6 month testing period.

The ambient environmental sensors accurately measure temperature, humidity, and pressure and are important for correcting the environmentally related offset in electrochemical sensor readings. The TE Connectivity MS5540C is a barometric pressure

sensor capable of measuring across a 10 to 1100 mbar range with 0.1 mbar resolution. Across 0 C to 50 C, the sensor is accurate to within 1 mbar and has a typical drift of +/- 1 mbar per year. The Sensiron SHT11 is a relative humidity sensor capable of measuring across the full range of relative humidity (0 to 100% RH) with a 0.05% RH resolution. Both sensors come equipped with temperature sensors with $\pm 0.8$ C and $\pm 0.4$ C accuracy, respectively. The sensors stabilize to environmental changes in under 30 seconds, which is sufficiently fast to accurately capture changes in the local environment.

In order to improve the robustness of the boards to ambient conditions, the electronics were conformally coated with silicone and placed into an enclosure as shown in Figure 3. The housing prevents direct contact with the sensors by providing ports over the electrochemical sensors and a vent near the ambient environmental sensors. The system relies on passive diffusion of pollutants into the sensors due to the high power cost of active ventilation. However, as described in Section 2.3, for this study the housed sensor packages were placed in an actively ventilated container.

**2.2.2  Software Infrastructure**

We developed two applications for Android smartphones that leverage the BLE connection of the MetaSense platform. The first application, the MetaSense Configurator app, enables users to configure the hardware for particular deployment scenarios, adjusting aspects such as sensing frequency, power gating of specific sensors connected, and the communication networks utilized. The second application, simply called the MetaSense app, collects data from the sensor via BLE and uploads all



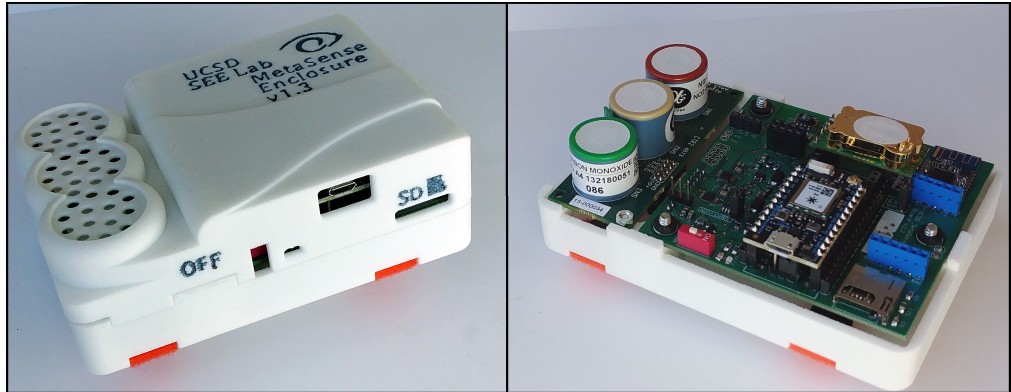

**Figure 3.** An enclosure was 3D printed for the MetaSense Air Quality Sensing Platform with top-side ports above the electrochemical sensors and a side port next to the ambient environmental sensors.

readings to a remote database. Each sensor reading is stamped with time and location information, supporting data analysis for mobile use cases. Moreover, users can read the current air quality information on their device, giving them immediate and personalized insight into their exposure to pollutants.

The remote measurements database is supported by the MetaSense cloud application and built on Amazon's AWS cloud.
Not only can the MetaSense app connect to this cloud, but the MetaSense boards can be configured to connect directly to it using Wi-Fi or 3G. The measurement data can be processed by machine learning algorithms in virtual machines in AWS or the data can be downloaded to be analyzed offline. The aforementioned over-the-air firmware updates are handled through Particle's cloud, which also allows remotely monitoring, configuring and resetting boards. These direct-to-cloud features are key to supporting a long-term, wide-scale deployment like the one presented in this paper.

**2.3 Data Collection and Preprocessing**

To support a long-term deployment in potentially harsh conditions, the sensors were placed into environmentally robust containers, shown in Figure 1, bottom right. The container was a dry box, measuring 27.4 x 25.1 x 12.4 cm, that was machined to have two sets of two vents on opposing walls. Louvers were installed with two 5 V, 50 mm square axial fans expelling ambient air from one wall and two louvers allowing air to enter the opposite side. The configuration allowed the robust container to
15 equilibrate with the local environment for accurate measurement. Each container could hold up to three MetaSense boards with cases and complementary hardware. Due to the long timeframe of the deployment, a USB charging hub was installed into the container to power the fans, the air quality sensors, and either a BLU Android phone or Wi-Fi cellular hotspot. The phones and hot spots were used to connect the sensors to the cloud; therefore, we could remotely monitor the sensors' status in real-time and perform preliminary data analysis and storage. Each board also had an SD card to record all measurements
locally, increasing the reliability of data storage.



**Table 1.** Board locations and dates for each round.

|          | Round 1<br>9/26/17 - 10/19/17 | Round 2<br>10/19/17 - 12/21/17 | Round 3<br>12/21/17 - 3/5/18 |
|----------|-------------------------------|--------------------------------|------------------------------|
| **Board 17** | El Cajon | Shafter  | Donovan  |
| **Board 19** | El Cajon | Shafter  | Donovan  |
| **Board 21** | El Cajon | Shafter  | Donovan  |
| **Board 11** | Shafter  | Donovan  | El Cajon |
| **Board 12** | Shafter  | Donovan  | El Cajon |
| **Board 13** | Shafter  | Donovan  | El Cajon |
| **Board 15** | Donovan  | El Cajon | Shafter  |
| **Board 18** | Donovan  | El Cajon | Shafter  |
| **Board 20** | Donovan  | El Cajon | Shafter  |

Each container holding three MetaSense sensor packages was placed at one of three sites for simultaneous data collection across the sites. After a period of time the containers were rotated to a new site such that every package spent a period of time at every site. We performed three rotations such that every sensor was returned to its original site for the final collection, but we disregarded the data from the initial round 0, except to verify that sensor performance had not changed measurably between

the beginning and the end of the deployments. Table 1 lists the dates for each rotation as well as where each sensor system was located for each rotation. The dates are approximate due to the logistics of gaining access to regulatory field sites and the distances traveled to deploy sensors. Also of note is that the deployments are not of equal length. This does not affect the results reported below because we ran all combinations of training and testing sites, and training set sizes were normalized to remove the influence of training set size. The data from the reference monitors was provided by the cooperating air quality districts in

the form of minute-averaged $O_3$ and $NO_2$ concentrations for the time period that our sensor packages were deployed.

Prior to using the dataset for training the calibration models, we performed a preprocessing step. First, we programmatically filtered out data samples that contained anomalous values that might have occurred due to a temporary sensor board malfunction (e.g., due to condensation). Specifically, we searched for temperature and voltage spikes that were outside the realm of reasonable values (i.e., temperature values above 60 degrees Celsius or ADC readings above 5 volts) and removed

the corresponding measurements. Each removed sample was visually inspected to ensure data was not being erroneously removed. The remaining data was averaged over a minute window to match the time resolution of the data from the reference monitors. Although we gathered sensor voltage measurements from both the auxilliary ($AE$) and working electrodes ($WE$) of the electrochemical sensors, we used the difference between the two ($AE - WE$) as the representative voltage for each sensor since the auxilliary voltage is meant to serve as a reference voltage for the working electrode. This treatment is consistent with

the methodology of Zimmerman et al. (2018), and we validated that the performance of the calibration models did not differ between tests with both electrodes and test with the difference as input features. The resulting data set over the three rounds at the three site contains 1,200,000 minute-averaged measurements.





With this data, we were able to verify our hypothesis in Section 2.1 that we would observe varied environmental and pollutant conditions among the sites. Generally higher ozone values were reported at Shafter, whereas generally higher $NO_2$ values were reported at Donovan. Higher humidity values were reported at the Donovan and El Cajon sites, as compared to Shafter. Some of the lowest temperature values were reported at Shafter. For more information see the distribution plots in Appendix A.

## 2.4 Baseline Calibration Methods

Sensor calibration is the process of developing and training models to convert a sensor voltage into a pollutant concentration. We formulate sensor calibration as a regression problem with input features $x$ and $e$ representing signals from the electrochemical sensors ($O_3$ voltage, $NO_2$ voltage, CO voltage) and environmental factors (temperature, pressure, humidity), respectively, for a total of 6 features. These features are input to a calibration function $h_\theta(x,e)$ that estimates target values $y$ representing pollutant concentrations ($O_3$ ppb and $NO_2$ ppb).

In our regression problem, we seek a function such that $h_\theta(x,e) \approx y$, which we formulate as an optimization where we minimize error over a training data set $\{x_n, e_n, y_n\}_{n=1}^{N}$ according to a loss function $L(h_\theta(x,e), y)$, i.e.

$$\theta^* = \arg\min_\theta \frac{1}{N} \sum_{n=1}^{N} L(h_\theta(x_n, e_n), y_n) \tag{1}$$

Models trained in this way assume that at inference time, predictions are made on data sampled from the training distribution. While this assumption holds true when the air quality sensors are trained and tested at the same site, the distribution of pollutants and environmental conditions changes when the sensors are moved to a new location.

We investigated the performance of three calibration models: multiple linear regression, neural networks (sometimes called deep learning), and random forest. These methods vary in their ability to accurately model complex behaviors, otherwise known as *capacity*, with linear regression having relatively low capacity and neural nets and random forests having substantial capacity. The price of high capacity is the potential to overfit the training distribution, which is a failure to generalize beyond the training data. Models that overfit will incur significant error when predicting on out-of-distribution examples. Overfitting can be mitigated with regularization and by reducing the model capacity, but this can only go so far if the testing distribution is substantially different from the train distribution. All of these methods have been previously applied to ambient pollutant estimation by various research groups (Piedrahita et al., 2014; Spinelle et al., 2015b, 2017; Sadighi et al., 2018; Zimmerman et al., 2018; Casey and Hannigan, 2018) and are generally common predictive modeling methods. For neural nets, we investigated three variants: two-layer, four-layer, and four-layer with a "split" architecture, which we motivate and describe in the next subsection.

Our baseline models were trained using the Scikit-Learn Python package, and the model parameters for each baseline model can be seen below:

1. **Linear regression:** we assume the functional form $h(x) \triangleq w^T x + b$, and fit the parameters in closed form. We use no regularization or polynomial features.





2. **Two-layer neural network:** we fit a two-hidden layer (200 wide) multilayer perceptron with rectified-linear unit activation functions and a final linear layer. We train this neural network using the Adam optimizer ($\beta_0 = 0.9, \beta_1 = 0.999$) and a learning rate of $10^{-3}$.

3. **Four-layer neural network:** Same as two-layer neural network, but four hidden layers of width 200 instead of two.

5    4. **Random forest:** We divide our data into five folds and train a random forest of size 100 on each fold, resulting in 500 trees. We aim to reproduce the strategy of Zimmerman et al. (2018) as closely as possible.

## 2.5 Split Neural Network Method

Overfitting is a problem for high capacity models with a limited distribution in training data, resulting in poor performance when a model is transferred to new locations and environments. One method to improve model transferability would be to collect more training data that includes the test distribution. However, colocating a sensor at multiple different regulatory field sites in order to capture a sufficiently wide distribution is prohibitive in terms of cost and time. An alternative solution is to deploy a set of sensors based on the same technology across multiple sites and then pool their data. However, there can be substantial sensor-to-sensor variance in performance that would amplify prediction errors. We propose a training architecture that consists of two sets of models: a global calibration model that leverages the data from a set of similar sensors spread across different training environments and sensor-specific calibration models that detect and correct the differences between sensors.

In the previous subsection, we associated each board $i$ with a calibration function $h_{\theta_i}(x)$ and fit this calibration function with its colocated data. Taking into consideration a collection of many air quality sensors, we propose an alternate architecture based on transfer learning (Goodfellow et al., 2016, p. 535). We propose using a calibration function split into two distinct steps: first, pollutant sensor voltages $x$ are input into a sensor-specific model, $s_{\theta_i}(x)$, a function parameterized by $\theta_i$, which outputs a fixed dimensional vector $u$. This intermediate representation $u$ is concatenated with environmental data $e$, which is then passed into a global calibration model $c_\phi([u|e])$. For a single air quality sensor, our final calibration function is $c_\phi([s_{\theta_i}(x)|e])$. Figure 4 depicts the use of such a model. Such a model is called a split neural network model (*split-NN*) since neural networks are generally used for both the sensor-specific models and the global calibration models. In our experiments, the sensor-specific model $s_{\theta_i}$ is either a linear regressor or neural network and $c_\phi$ is a two-layer, 100-wide neural network.

The purpose of the split-NN model is that $s_{\theta_i}$ corrects for differences in air quality sensor $i$'s performance relative to the other sensors, thus normalizing the values and making the behavior of all the sensors compatible with the global model $c_\phi$. The performance of the estimates from $c_\phi$ should be superior to those from an individual sensor model because it has been trained on the (normalized) data of all the boards as opposed to just a single board.

The split model can be trained efficiently with stochastic gradient descent. Specifically, we first collect $N$ data sets for each board $D_i = \{x^{(i)}, e^{(i)}, y^{(i)}\}_{i=1}^N$. We ensure each of these data sets is the same size by sampling each with replacement to artificially match the largest data set. We then pool the data sets together into one data set from which we sample mini-batches. While each sensor-specific model $s_{\theta_i}$ is trained only on data collected by its sensor, the regression with the other $s_{\theta_i}$ sensor-




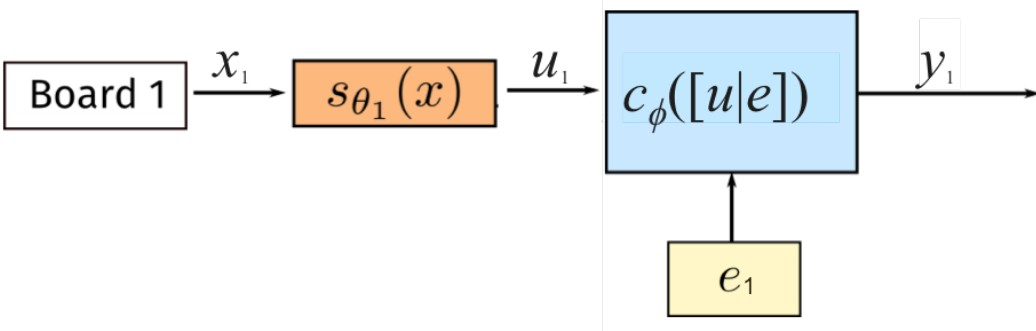

**Figure 4.** Architecture of the split-NN model in deployment (testing). Each air quality sensor has a board-specific model $s_{\theta_i}(x)$ that normalizes a given sensor's output ($x$) to an intermediate representation from all sensors ($u$). The intermediate representation is combined with environmental data ($e$) and input to the global model $c_\phi$.

specific models is designed to detect and correct its bias, outputting an intermediate representations $u$ that is normalized with the others. The global calibration model is trained on the normalized data from all air quality sensors.

Although training this neural network will take longer than training one for a single board, it has several key advantages over conventional calibration techniques. The first is its ability to share information across multiple boards. Suppose Board A is trained on Location 1 and Board B is trained on Location 2. Pooling the data sets and using a shared model enables the global calibration model to predict well in both locations, and the calibration models for both boards will have information about the other locations in them, in theory improving transferability. The second is more efficient utilization of data. By pooling data and training jointly, we effectively multiply our data size by the number of boards. Alternatively, field deployments can be shortened.

**Calibrating a New Board without a Full Training.** Field calibration is traditionally performed by colocating a sensor package with reference monitors and then training to match pollutant concentrations. But, suppose we already had a fleet of low-cost sensor packages already deployed. A simpler method not requiring coordination with regulatory agencies would be to colocate it with a calibrated sensor package and train a model to match its predicted pollutant levels. This risks compounding errors across models, however.

The split-NN model enables calibrating a new sensor package by colocating to match *representation* instead of predictions, as learned representations can often improve generalization in transfer learning problems (Goodfellow et al., 2016, p. 536). We propose calibrating sensor package $N+1$ to match the intermediate representation output of a colocated, previously-calibrated sensor package. Specifically, we train model $N+1$ to minimize $L(u_N, u_{N+1})$, or the loss between the two packages' intermediary outputs. These intermediate representations are designed to be robust to changes in location so training to match these representation so it is expected that it will result in a robust calibration model. We analyze this potential calibration technique by holding out a board from our data sets and training a split model. We then simulate calibrating the held out board





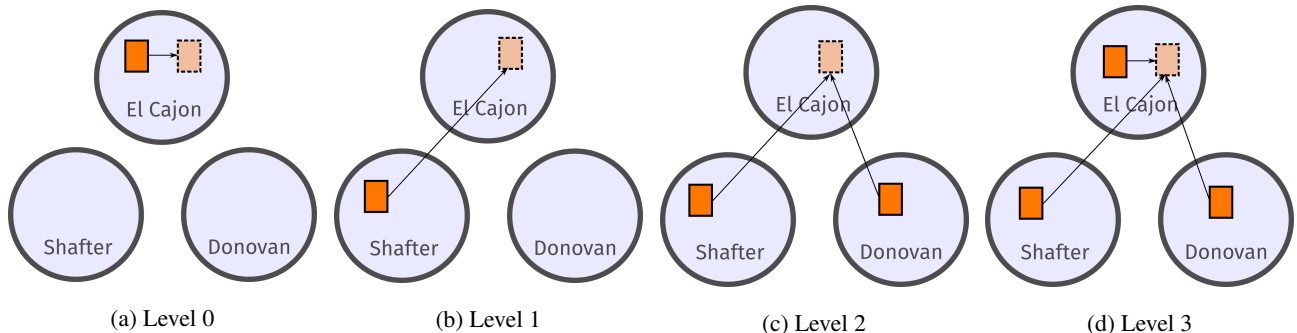

(a) Level 0          (b) Level 1          (c) Level 2          (d) Level 3

**Figure 5.** Graphical depiction of training versus testing for the Level 0 through Level 3 benchmarks. The Level 0 and 3 benchmarks test on a training site using held out data. The Level 1 and 2 benchmarks train and test on different sites, also using held out data for consistency.

by training a sensor model to match the representations produced by another board it was colocated with. We then use this new sensor model with the global calibration function to produce pollutant values.

## 3   Results and Discussion

### 3.1   Robustness of Different Calibration Techniques Across New Locations

We evaluated a set of four baseline models described in Section 2.4: multiple linear regression, two-layer neural network (NN-2), four-layer neural network (NN-4), and random forest (RF). With each of these four models, we performed a suite of identical calibration benchmarks that measure the robustness of models to out-of-distribution data. We split all data sets uniformly at random into training and testing subsets, reserving 20% of each board's data for testing. In each benchmark, we progressively widened the training distribution by combining training data from more locations (using subsampling to maintain the training
set size), while keeping the testing set data set from one location. We have four "levels" of such benchmarks:

– **Level 0:** Train a model on one location and test on the same location. Several studies, discussed in Section 1, have previously assessed this configuration (Zimmerman et al., 2018; Spinelle et al., 2015b, 2017; Cross et al., 2017).

– **Level 1:** Train a model on one location and test on another location. Some recent studies, also discussed in Section 1, have previously studied this configuration (Hagan et al., 2018; Casey and Hannigan, 2018; Bigi et al., 2018; Malings
et al., 2018).

– **Level 2:** Train a model on two locations and test on a third location.

– **Level 3:** Train a model on three locations and test on one of the three locations.

In the Level 0 and Level 3 benchmarks, the training and testing data distributions have explicit overlap, whereas in Level 1 and 2, there is no explicit overlap. We expect performance on Level 0 to be the best, as the training and testing distributions are
identical. We expect performance on Level 3 to be similar, due to the overlap in training and testing distributions. We expect





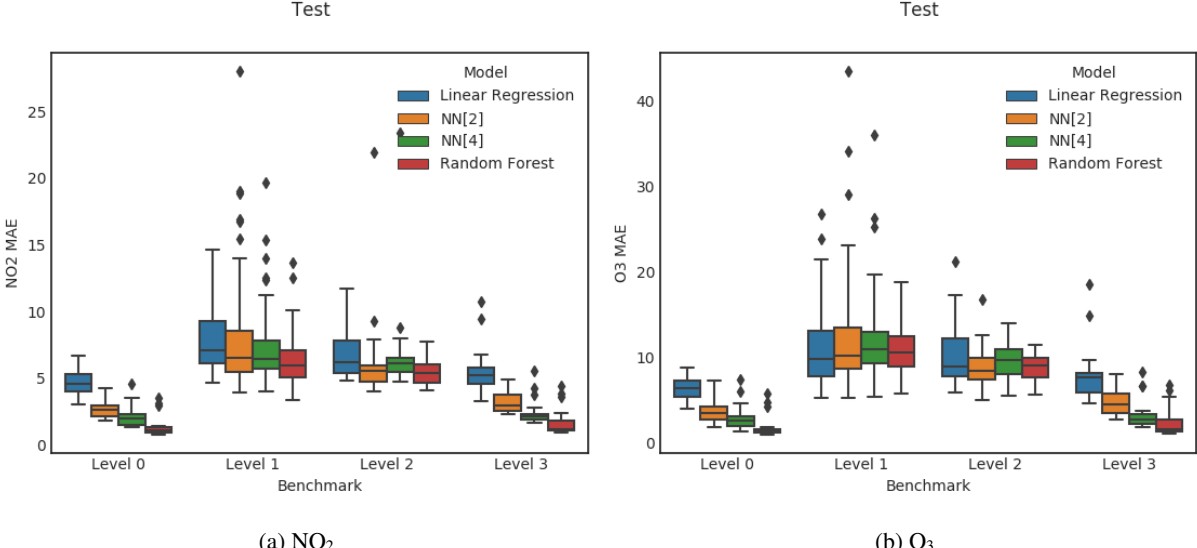

(a) NO$_2$            (b) O$_3$

**Figure 6.** Mean absolute error (MAE) boxplots for NO$_2$ and O$_3$, for the Level 0 through Level 3 benchmarks.

performance on Level 1 to be the worst, as the training distribution is the narrowest and with no explicit overlap, whereas we expect performance on Level 2 to be between Level 1 and Level 3, for although there is no explicit overlap, the overall training distribution will be wider, forcing the models to be more general and possibly affording more implicit overlap. Furthermore, we expect higher capacity models to overfit more to the training data set, and as a result, have the largest gap between Level 0

and Level 1. Thus, we expect linear regression to have more consistent performance across the benchmarks, albeit at relatively high error, followed by the 2-layer neural network, 4-layer neural network, and finally the random forest.

    We ran each benchmark across all possible permutations of location and sensor package, measuring six metrics: root mean squared error (rMSE), centered root mean squared error (crMSE), mean absolute error (MAE), the coefficient of variation of mean absolute error (CvMAE), mean bias error (MBE), and coefficient of determination ($R^2$). The results for MAE of the

baseline models are plotted in Figure 6. Details can be explored further in Appendix C.

    We observe that on average, as model capacity increases, Level 0 error decreases. This is consistent across both NO$_2$ and O$_3$ prediction and reflects the ability of the model to fit the training distribution. Concerning model transferability, we find that consistently, *all models suffer significant error when tested on different locations*. Level 1 and 2 benchmarks reflect the ability of a model to generalize to a distribution it hasn't seen before and we see that in these benchmarks, errors are much higher

and the gaps between models are much smaller. Furthermore, Level 2 error is slightly lower on average than Level 1 error. By adding data from another site, effectively widening the training distribution, the models are slightly more robust to the unseen testing distribution. Level 3 performance aligns closely with Level 0 performance, which is to be expected, since in both cases the training distribution contains the testing distribution.

    Across baselines, we observe that on average, linear regression has the highest error on all the benchmarks. However, its

errors across the Level benchmarks are more consistent than the other models, suggesting that low-capacity linear regression




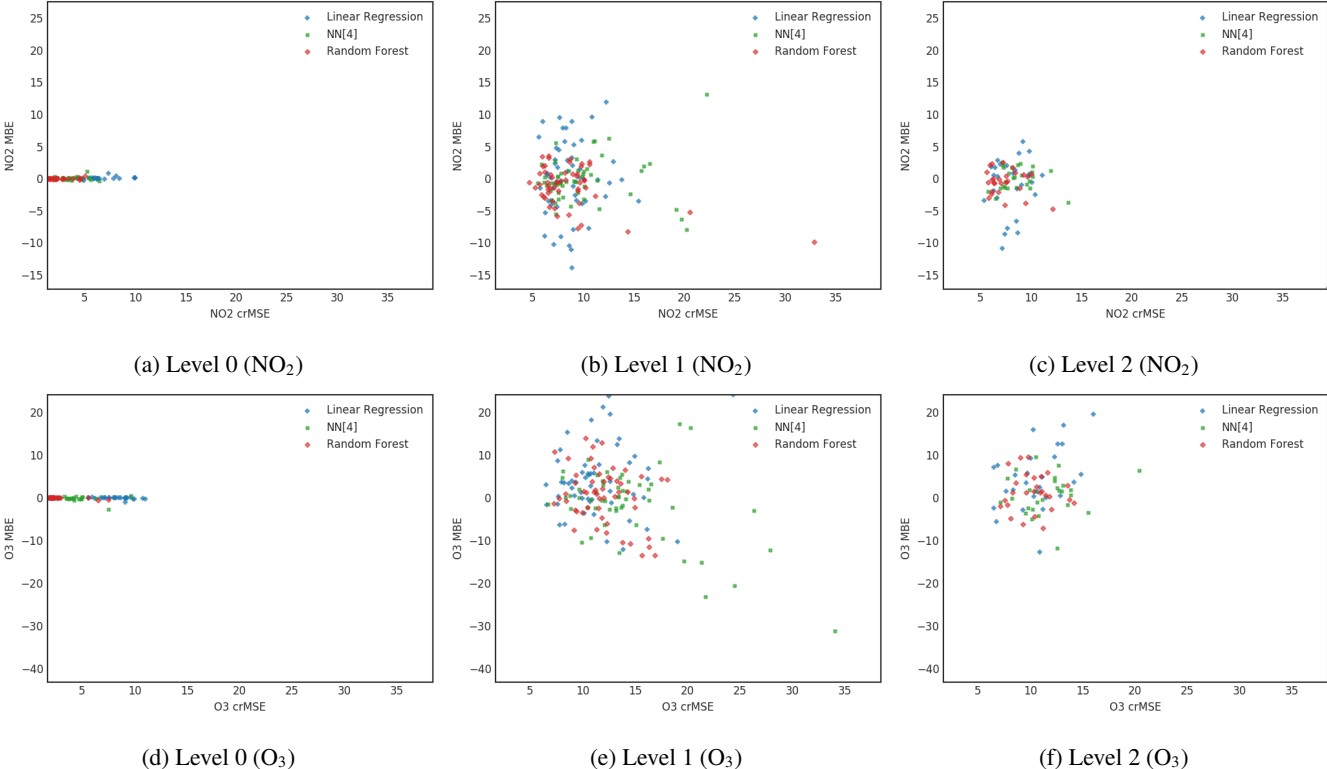

**Figure 7.** Target plots for Level 0 through Level 2 for both NO$_2$ and O$_3$.

is more robust to transfer. On the other hand, random forests have on average the lowest error, but have the most inconsistent results across the Levels. The results indicate a tradeoff between model capacity and robustness to transfer, consistent with our intuitions about model overfitting and generalization. Neural networks lie in between linear regression and random forests, and offer a tradeoff between low error and consistent error.

To better understand how model performance degrades, we produced target plots, which visualize the tradeoff between centered error and bias error (Figure 7). The target plots indicate that while error approximately doubles when there is no explicit overlap in the distribution, model bias is many times larger. The increase in bias is more pronounced in the higher capacity models. Furthermore, despite the higher capacity models showing better error and bias in a Level 0 benchmark, the models have very similar error-bias tradeoffs in a Level 1 benchmark, indicating that even a high-capacity model cannot avoid

this performance degradation. Finally, in comparing the Level 1 and Level 2 plots, we observe that adding an additional (no-overlapping) site primarily reduces bias. The Level 3 plots are very similar to the Level 0 plots and are excluded from Figure 7 for brevity.

In general, however, we observe that model performance degrades non-trivially when moved to different locations. This decrease in performance could result in overconfidence in a sensor's readings, potentially affecting downstream decisions. We

briefly analyze the properties of our data that could result in overfitting by first investigating how data distributions across



sites and times differ. Over each location and round, pollutant values can be highly variable. This is reflected, for example, in Figure A3 where Shafter has higher values of $NO_2$ in Round 1 and 2 but lower in Round 3. Furthermore, in Figure A4, the distribution of $O_3$ changes remarkably across round and location. Similarly, temperature and humidity change significantly across location and round, which can be seen in Figure A1 and Figure A2.

A question that remains is to what degree overfitting or unique (non-overlapping) distributions of environmental data at the sites is contributing to the failure of the high capacity models to transfer well. In an effort to better understand what may be driving the drop in performance of the high capacity models when boards are moved, we examined error density plots for temperature and humidity for the Level 1 benchmarks. In these types of plots, one of the predictors, such as temperature or humidity, is plotted against the error for all three sites in a single plot. Figure 8 displays the error density plots for absolute

humidity against the error for the $O_3$ estimation, for both the linear regression and random forest models. These plots illustrate how the magnitude of error varies with respect to higher or lower predictor values as well as how different pairs of training and testing sites compare. There are a couple of things we can derive from this collection of plots. First, we observe that the pollutant concentrations at the Shafter site are difficult to predict, except for random forest when trained at Shafter itself (Figure 8f). The Shafter site was spatially far from the other sites and likely had a unique composition of background pollutants

and ambient environmental conditions. Second, we observe that when training a random forest model at one site and testing it at a different site (Figure 8, bottom row), the error density plots look similar to the results from the linear regression models (Figure 8, top row) despite the higher capacity of random forest models. Furthermore, comparing panels a and d, the errors at Shafter seem comparable to those at El Cajon for the random forest model, whereas for the linear regression model the errors seem greater at Shafter versus the second San Diego site. A difference potentially indicating that linear regression models are

better at transferring between more similar environments, which has been observed by other researchers as well (Casey and Hannigan, 2018). We also observe that the greater errors at the Shafter site are occurring at humidity values that were seen in the training data set (more centrally in the plot), as is evident by their representation in the Donovan data. This implies that these errors did not occur at humidity values that have been extrapolated beyond the original training data set, but rather from overfitting at values in the distribution. This leads us to conclude that overfitting is the reason random forest's net performance

in transfer is not much better than linear regression.

## 3.2   Benefits of Sharing Data Across Sensor Packages

In this section, we evaluate the split-NN model architecture's utility for improving the transferability of a calibration model. The novelty of the split-NN model for calibrating a board's model is its ability include (normalized) data from other boards. Given that the resources for calibration are limited, the research questions for split-NN revolve around how boards could be

best distributed to available field sites. For a standard modeling technique like random forest, a board has to be placed at three sites for three rounds to experience the wide training distribution that achieves the exceptional transferability observed in the Level 3 benchmarks. However, with the split-NN model, multiple boards can be deployed for just one round, divided equally across the sites. Then the data from their boards can be normalized and shared to produce models that we hypothesize to be of similar quality to a Level 3 benchmark, but in one-third the time, in a single round.





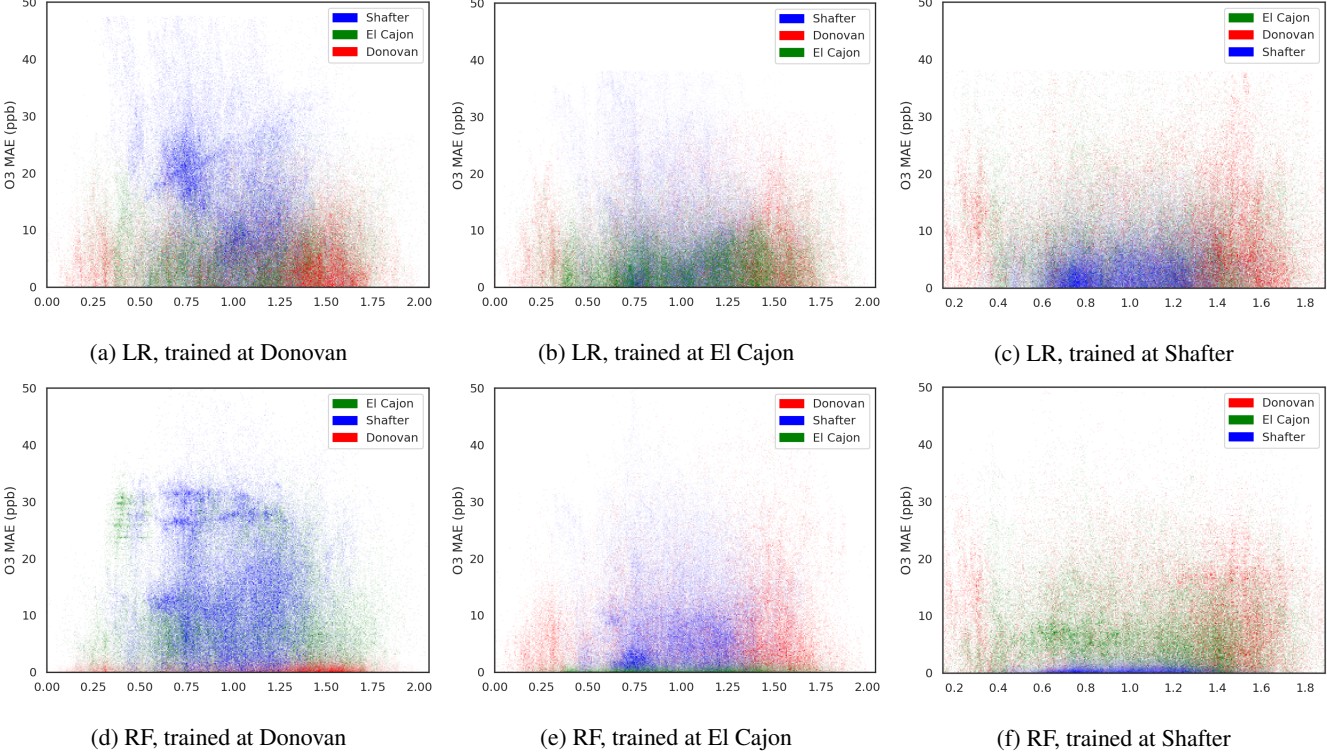

**Figure 8.** Error density plots for $O_3$ versus normalized absolute humidity for both Linear Regression (LR) and Random Forest (RF) in a Level 1 benchmark.

To help reveal the value of calibrating multiple boards at once, we performed three one-round benchmarks: 1 board at each of the three sites, 2 boards at each of the three sites, and 3 boards at each of the three sites. In each of these conditions, a board is trained from a single round of data and tested on the other locations, not its own. In this vein, these are all Level 1 benchmarks, thus we compare the resulting models against our Level 1 baselines. We expect the split-NN to outperform Level 1 random forest, as the inclusion of more data helps reduce bias. In the situation that there are more boards to calibrate than there are training sites, there is an opportunity to also incorporate data additional boards at the same site. We expect that a greater multiplicity of boards at each site will produce slightly better models, but with diminishing returns. We evaluated this effect by including training split-NN's with increasing numbers of boards at each site, indicated by the variants Split-NN (3), Split-NN (6), and Split-NN (9), corresponding to having one board at each site, two boards at each site, and three boards at each site. We perform a similar assessment with two-round (Level 2) benchmarks, still testing only on sites that a board hasn't been trained on. As previously, we control for the total amount of data, simulating an abbreviated deployment for the Level 2 benchmarks.

Figure 9a-b shows that the split-NN model on average has slightly lower MAE in the Level 1 benchmarks when compared to the random forest model. We see in and Figure 9c-d that the gap widens with the Level 2 benchmark, indicating that the



Split-NN model is able to better capitalize on the additional data. The results also support our hypothesis that we receive diminishing returns with additional data. Detailed results are provided in Appendix D.

The marginal improvement seen in the Level 1 benchmarks has two possible causes. One possibility is that the difference in behavior between sensors is non-linear. To test this, we implemented a full neural network as the first stage. The results

5   were comparable with a linear regression first stage with only slight improvement, suggesting that the relationship between the sensors is well represented by a linear model. The other possibility is that the pollution distributions have insufficient overlap across sites, compromising the first-stage linear regression to for correct bias. The fact that using two rounds of data (Level 2) does much better suggests that this lack of overlap is a likely culprit.

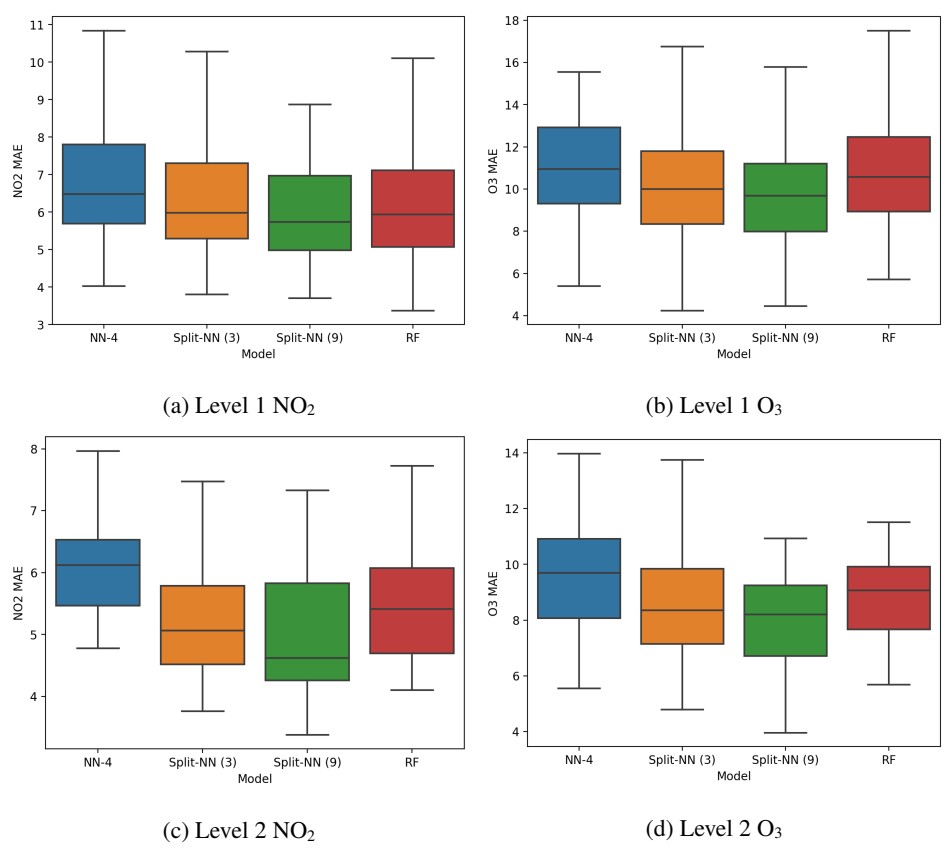

(a) Level 1 NO$_2$

(b) Level 1 O$_3$

(c) Level 2 NO$_2$

(d) Level 2 O$_3$

**Figure 9.** Results of evaluating the split-NN model with a linear regression first stage, compared against the RF model in both Level 1 and Level 2 comparisons. The split-NN model has a lower mean and median error in all conditions. Boxplots are pictured without outliers for clarity.

## 3.3   Discussion

10   As low-cost sensor studies move from understanding sensor signal performance to how this performance is affected by moving sensors to new sampling locations or utilizing them in new applications, it is important that the results are translated into



best practices to support the collection of usable high quality data. This is particularly important given the interest in sensors by community-based organizations and citizen scientists. Although the present study examined only electrochemical $O_3$ and $NO_2$ sensors and the sampling sites were limited to three in California, it adds to a body of evidence that location matters in the calibration of low-cost sensors because the background environmental conditions matter. With this in mind, we make the

following observations and recommendations.

We observed how prediction performance degrades when a sensor is moved to a new location, especially for high-capacity modeling techniques. In particular, training a complex random forest calibration model will likely result in very low error at a colocated site but can incur significant error at a different site. Although their predictions at a new site will have lower error than linear regression, the error they have at the training site will likely not be representative of their error in practice. A linear

model, on the other hand, despite not predicting as well at the training site, will not have significantly more error at testing time. Thus, if it is important to know the likely error of your calibration model under transfer, it would be best to use a low-capacity method like linear regression.

When we drilled down to investigate the contributors to error when changing location, we found that bias error was a significant contributor in many cases. This is interesting because bias error indicates a loss of accuracy (a non-random additive

error) rather than a loss of precision (random noise). This suggests that when moving a sensor to a new location, if the bias can somehow be detected, then it may be possible to make a bias correction to improve model performance. This result also motivates the use of the split neural network architecture, which has a model-specific correction stage that is designed to learn unbiased representations of sensor measurements.

We had expected that training at multiple sites would provide much better transferability, but the improvements were not

substantial, suggesting that the high-capacity models were mostly improving due to implicit overlap in distributions and not actual generalization. This suggests that calibration should be directed at capturing the widest conditions possible, for example using many field sites with varying conditions, so as to create an overlap between the distributions of training and use. This recommendation is further supported by the observation that the Level 3 benchmarks performed nearly as well as the Level 0 benchmarks, in spite of carrying the load of a much wider distribution in the models.

The split-NN approach provides a potentially economical approach to creating overlap in distributions since sensors can share their data for calibration. That is, when calibrating multiple sensors, rather than colocating multiple sensors at a field site and rotating those sensors over time, it makes sense to distribute the sensors to as many field sites as possible to capture the widest distribution of conditions. The split-NN method has the additional benefit of being able to train a calibration model for a sensor that has never been colocated with a reference instrument. By simply colocating an uncalibrated sensor with a calibrated

sensor and training the sensor-specific model to match the intermediate output of the calibrated sensor, the uncalibrated sensor can leverage the same global calibration model. More study will be required to see how well the split-NN approach scales as the training data distribution increases and to determine the bounds on calibration without reference colocation.



## 4   Conclusion

As low-cost gas-phase sensors are increasingly being adopted for citizen science efforts and community-based studies, there is a need to better understand what contributes to accurate sensing. A key question is how a change in background environmental or pollutant conditions, often unique to a location, affects accuracy. A rotating deployment strategy enabled benchmarking the

transferability of models and investigating how to improve accuracy. We found that overfitting is a concern, especially when transferring high-capacity models like random forest that are trained with data that will not be representative of the conditions of use. Our benchmarks indicate that widening the data distribution is a good strategy to make models more robust to transfer, but that the best results require the training distribution to contain the distribution encountered in use. A tantalizing result is that much of the error introduced by transfer was bias, which may be correctable. When multiple sensors based on the

same technology are being trained at the same time, we found that a split neural network architecture increases the robustness of model transfer by giving a sensor's model access to normalized data from other sensors, even at other locations, hence widening the distribution without requiring additional data collection. This method also enables accurately calibrating new sensors against existing calibrated sensors at incremental cost.

In the future work we will be extending this work to answer open questions that we believe are relevant to the future of

low-cost sensor calibration. As one example, there are questions about the effect of temporal resolution on accuracy. Currently, our MetaSense sensors are sampled every five seconds, but the ground-truth data provided from reference monitors is minute-averaged. By averaging our own sensor measurements every minute, we discard data that could be relevant for calibration. Recent advances in recurrent neural networks for sequence prediction might help leverage the high-resolution data for robust prediction. As a second example, a potential application of low-cost sensing is truly mobile sensing with person- or vehicle-

mounted sensors. Deployments such as these will raise questions about the effects of mobility on sensing accuracy, such as rapidly changing conditions, with few studies to date (Arfire et al., 2016).

## 5   Acknowledgements

Funding for this work was provided by the National Science Foundation (awards: CNS-1446912 and CNS-1446899) and KACST. We would also like to thank partners at the San Diego Air Pollution Control District as well as San Joaquin Valley

Air Pollution Control District for their support throughout this deployment.



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





## Appendix A:  Environment and Pollutant Distributions

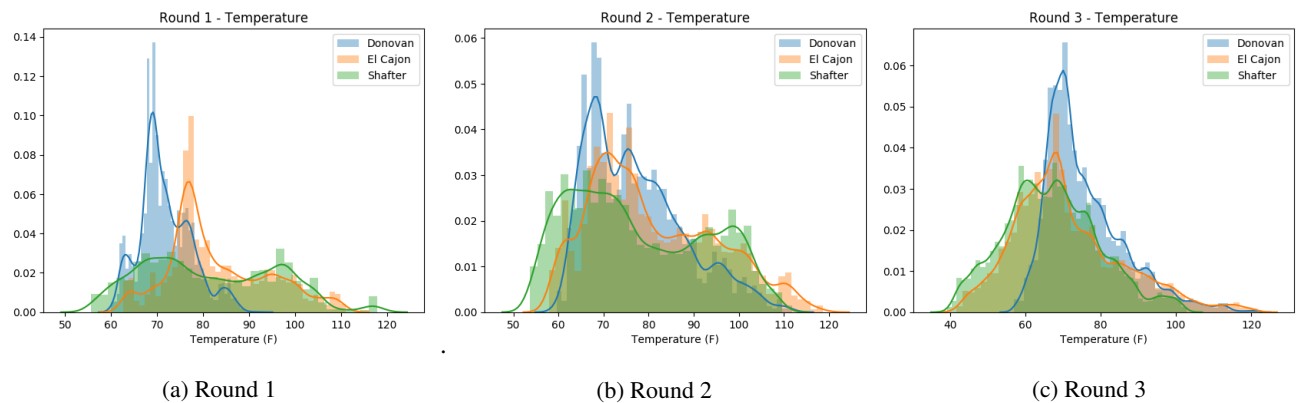

**Figure A1.** Temperature distributions for each location, by round.

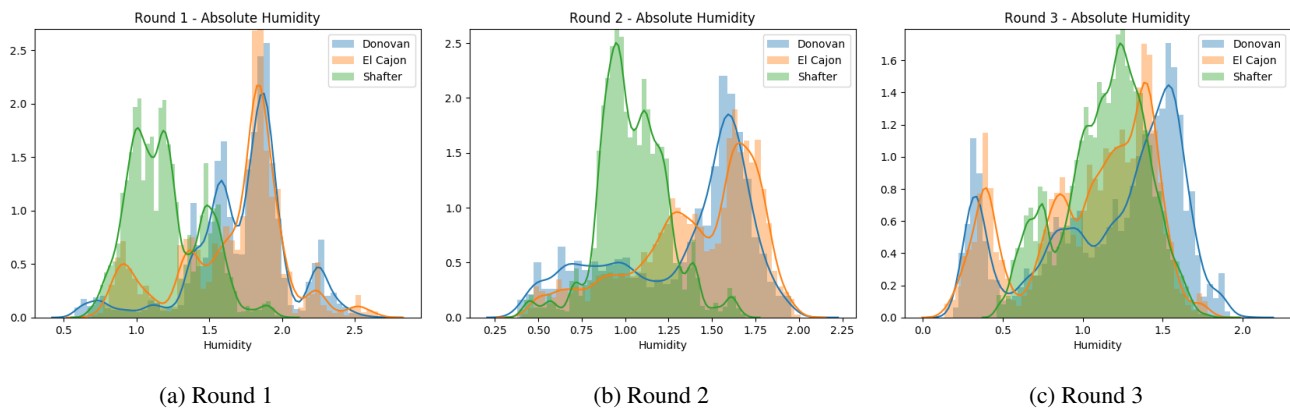

**Figure A2.** Humidity distributions for each location, by round.

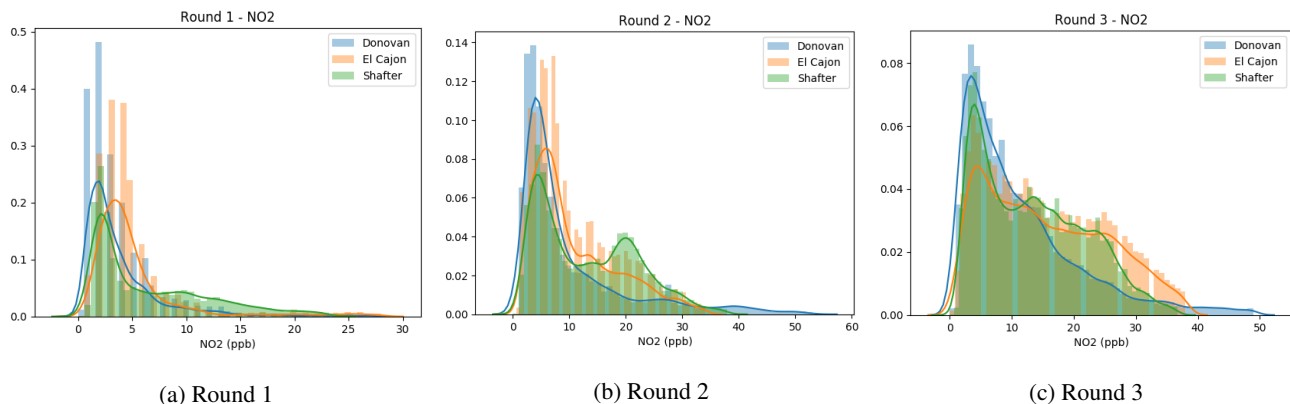

**Figure A3.** NO$_2$ distributions for each location, by round.





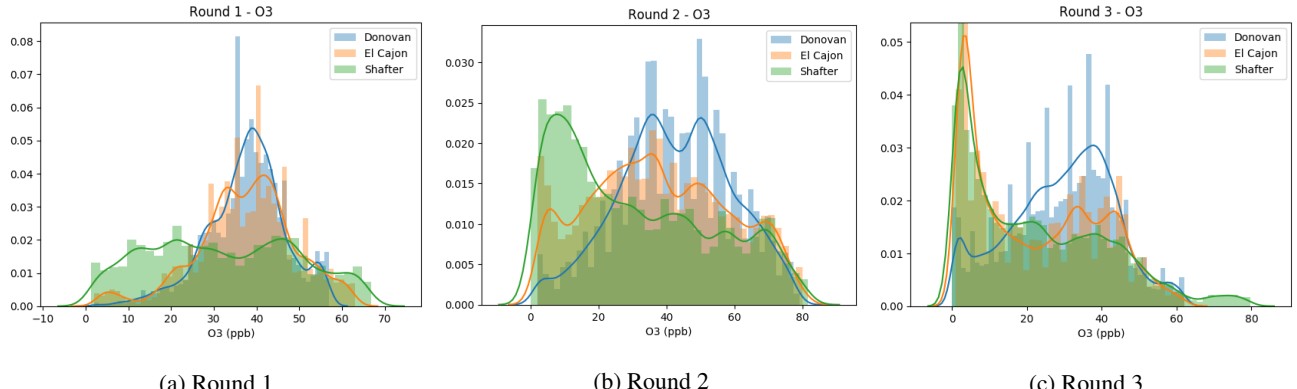

| (a) Round 1 | (b) Round 2 | (c) Round 3 |

**Figure A4.** O$_3$ distributions for each location, by round.

## Appendix B: Summaries of Data for each Location and Round

| Location | | epa-no2 | epa-o3 | temperature | pressure | humidity |
|---|---|---|---|---|---|---|
| Donovan | count | 100780.000000 | 100780.000000 | 100780.000000 | 100780.000000 | 100780.000000 |
| | mean | 10.433935 | 33.741754 | 24.169665 | 991.767096 | 45.936100 |
| | std | 10.806825 | 15.377884 | 5.623902 | 3.225683 | 21.965759 |
| | min | 0.000000 | 0.000000 | 13.900000 | 982.820000 | 4.086000 |
| | 25% | 3.000000 | 24.000000 | 20.100000 | 989.530000 | 27.244000 |
| | 50% | 7.000000 | 35.000000 | 22.620000 | 991.460000 | 49.511000 |
| | 75% | 14.000000 | 43.000000 | 27.000000 | 993.610000 | 64.394250 |
| | max | 157.000000 | 96.000000 | 49.710000 | 1004.160000 | 92.753000 |
| El Cajon | count | 97412.000000 | 97412.000000 | 97412.000000 | 97412.000000 | 97412.000000 |
| | mean | 12.914015 | 29.331499 | 24.341702 | 997.287606 | 43.923309 |
| | std | 9.732012 | 19.337040 | 8.232391 | 3.507203 | 20.076611 |
| | min | 0.000000 | 1.000000 | 5.430000 | 989.230000 | 2.733000 |
| | 25% | 5.000000 | 11.000000 | 18.570000 | 994.880000 | 28.623000 |
| | 50% | 10.000000 | 31.000000 | 23.380000 | 996.890000 | 45.052500 |
| | 75% | 20.000000 | 43.000000 | 29.700000 | 999.450000 | 61.166250 |
| | max | 66.000000 | 95.000000 | 49.790000 | 1010.480000 | 85.827000 |
| Shafter | count | 119785.000000 | 119785.000000 | 119785.000000 | 119785.000000 | 119785.000000 |
| | mean | 12.578259 | 26.357091 | 22.100842 | 1003.882723 | 45.804388 |
| | std | 9.078981 | 20.739128 | 8.184091 | 5.595797 | 18.072375 |
| | min | 0.000000 | 0.000000 | 4.010000 | 872.755556 | 6.349000 |
| | 25% | 4.700000 | 7.800000 | 16.155556 | 999.750000 | 30.585000 |
| | 50% | 10.800000 | 22.300000 | 21.040000 | 1003.990000 | 46.763000 |
| | 75% | 19.000000 | 41.200000 | 27.200000 | 1007.400000 | 60.965000 |
| | max | 594.600000 | 110.400000 | 47.700000 | 1019.580000 | 85.047000 |

**Table B.1.** Summary of data set grouped by location





| Round | | epa-no2 | epa-o3 | temperature | pressure | humidity |
|---|---|---|---|---|---|---|
| 1 | count | 49771.000000 | 49771.000000 | 49771.000000 | 49771.000000 | 49771.000000 |
| | mean | 5.508949 | 36.009789 | 26.061956 | 994.458846 | 48.321718 |
| | std | 5.472207 | 13.868935 | 6.776979 | 4.786601 | 19.539286 |
| | min | 0.000000 | 1.300000 | 13.100000 | 872.755556 | 9.644000 |
| | 25% | 2.000000 | 28.000000 | 20.900000 | 990.920000 | 31.790000 |
| | 50% | 3.700000 | 37.300000 | 24.600000 | 995.240000 | 50.507000 |
| | 75% | 6.600000 | 45.000000 | 30.130000 | 997.640000 | 61.524500 |
| | max | 57.000000 | 110.400000 | 47.700000 | 1002.940000 | 92.753000 |
| 2 | count | 75129.000000 | 75129.000000 | 75129.000000 | 75129.000000 | 75129.000000 |
| | mean | 11.915902 | 36.973784 | 25.952835 | 995.989062 | 41.511134 |
| | std | 9.583055 | 21.259252 | 7.576508 | 6.074511 | 19.756994 |
| | min | 0.000000 | 0.000000 | 12.000000 | 982.820000 | 4.420000 |
| | 25% | 5.000000 | 19.200000 | 20.000000 | 990.990000 | 23.461000 |
| | 50% | 8.000000 | 36.000000 | 24.400000 | 995.420000 | 41.539000 |
| | 75% | 17.900000 | 53.000000 | 31.710000 | 1000.710000 | 56.961000 |
| | max | 82.000000 | 96.000000 | 48.180000 | 1009.890000 | 87.562000 |
| 3 | count | 193077.000000 | 193077.000000 | 193077.000000 | 193077.000000 | 193077.000000 |
| | mean | 13.708433 | 25.092950 | 21.791316 | 999.732162 | 45.945741 |
| | std | 10.225202 | 17.802680 | 7.275538 | 6.707979 | 20.012530 |
| | min | 0.000000 | 0.000000 | 4.010000 | 986.770000 | 2.733000 |
| | 25% | 5.100000 | 8.000000 | 17.190000 | 994.300000 | 30.192000 |
| | 50% | 11.600000 | 24.700000 | 21.000000 | 999.090000 | 48.470000 |
| | 75% | 20.000000 | 38.500000 | 25.780000 | 1004.690000 | 63.450000 |
| | max | 594.600000 | 87.900000 | 49.790000 | 1019.580000 | 85.440000 |

**Table B.2.** Summary of data set grouped by round

## Appendix C: Raw Results for the Baseline Calibration Models

The following tables are the complete error results for the baseline models across the various conditions. In these tables, the modeling methods are labeled as MLR for multiple linear regression, NN-2 for 2-layer neural network, NN-4 for 4-layer neural network, and RF for random forest, as described in Section 2.4. Likewise, the error measures are labeled as MAE for mean absolute error, CvMAE for coefficient of variation of the mean absolute error, MBE for mean bias error, MSE for mean standard error, $R^2$ is the coefficient of determination, crMSE for centered root mean square error, and rMSE for root mean squared error. The results are disaggregated by train and test sites, and averaged across the sensor packages.



| Train Site | Method | MAE | CvMAE | MBE | MSE | R^2 | crMSE | rMSE |
|---|---|---|---|---|---|---|---|---|
| elcajon | MLR | 3.860 | 0.256 | 1.163e-14 | 28.094 | 0.685 | 5.259 | 5.259 |
| donovan | MLR | 5.520 | 0.567 | 1.528e-15 | 73.035 | 0.312 | 8.374 | 8.374 |
| shafter | MLR | 4.671 | 0.354 | 3.628e-15 | 40.945 | 0.492 | 6.380 | 6.380 |
| elcajon | NN-2 | 2.003 | 0.135 | 0.127 | 8.137 | 0.905 | 2.831 | 2.844 |
| donovan | NN-2 | 3.134 | 0.328 | 0.093 | 27.100 | 0.733 | 5.175 | 5.189 |
| shafter | NN-2 | 2.648 | 0.200 | 0.051 | 17.439 | 0.787 | 4.131 | 4.135 |
| elcajon | NN-4 | 1.109 | 0.074 | 0.076 | 2.976 | 0.967 | 1.700 | 1.704 |
| donovan | NN-4 | 1.946 | 0.213 | 0.033 | 13.955 | 0.835 | 3.527 | 3.548 |
| shafter | NN-4 | 1.755 | 0.133 | -0.054 | 8.541 | 0.895 | 2.868 | 2.872 |
| elcajon | RF | 0.477 | 0.032 | -0.011 | 0.673 | 0.993 | 0.808 | 0.808 |
| donovan | RF | 0.999 | 0.112 | -0.022 | 3.705 | 0.956 | 1.870 | 1.871 |
| shafter | RF | 0.514 | 0.039 | -0.016 | 1.513 | 0.981 | 1.193 | 1.193 |

**Table C.1.** Level 0 train results for NO2 (train and test on the same data set).

| Train Site | Test Site | Method | MAE | CvMAE | MBE | MSE | R^2 | crMSE | rMSE |
|---|---|---|---|---|---|---|---|---|---|
| elcajon | elcajon | MLR | 3.869 | 0.256 | 0.015 | 28.999 | 0.683 | 5.333 | 5.333 |
| donovan | donovan | MLR | 5.573 | 0.556 | 0.160 | 75.037 | 0.315 | 8.504 | 8.512 |
| shafter | shafter | MLR | 4.671 | 0.353 | 0.004 | 38.467 | 0.511 | 6.191 | 6.191 |
| elcajon | elcajon | NN-2 | 2.074 | 0.140 | 0.127 | 9.185 | 0.895 | 3.006 | 3.018 |
| donovan | donovan | NN-2 | 3.390 | 0.345 | 0.204 | 31.458 | 0.696 | 5.559 | 5.581 |
| shafter | shafter | NN-2 | 2.694 | 0.203 | 0.052 | 15.772 | 0.802 | 3.942 | 3.946 |
| elcajon | elcajon | NN-4 | 1.465 | 0.098 | 0.076 | 5.524 | 0.938 | 2.327 | 2.331 |
| donovan | donovan | NN-4 | 2.739 | 0.288 | 0.105 | 24.875 | 0.729 | 4.907 | 4.924 |
| shafter | shafter | NN-4 | 2.089 | 0.158 | -0.060 | 10.702 | 0.865 | 3.252 | 3.256 |
| elcajon | elcajon | RF | 0.972 | 0.064 | -0.028 | 2.929 | 0.968 | 1.683 | 1.683 |
| donovan | donovan | RF | 2.010 | 0.216 | 0.031 | 15.113 | 0.830 | 3.794 | 3.797 |
| shafter | shafter | RF | 1.028 | 0.078 | -0.041 | 3.822 | 0.951 | 1.943 | 1.943 |

**Table C.2.** Level 0 test results for NO2.




| Train Site | Method | MAE | CvMAE | MBE | MSE | R^2 | crMSE | rMSE |
|---|---|---|---|---|---|---|---|---|
| elcajon | MLR | 6.010 | 0.245 | 1.647e-14 | 60.276 | 0.827 | 7.666 | 7.666 |
| donovan | MLR | 6.916 | 0.193 | -1.881e-15 | 85.486 | 0.584 | 9.131 | 9.131 |
| shafter | MLR | 5.882 | 0.239 | 9.460e-15 | 63.567 | 0.841 | 7.877 | 7.877 |
| elcajon | NN-2 | 2.810 | 0.112 | -0.150 | 16.200 | 0.954 | 3.940 | 3.947 |
| donovan | NN-2 | 4.237 | 0.117 | -0.270 | 35.055 | 0.821 | 5.824 | 5.855 |
| shafter | NN-2 | 3.498 | 0.141 | 0.013 | 24.929 | 0.939 | 4.895 | 4.909 |
| elcajon | NN-4 | 1.369 | 0.055 | -0.092 | 4.418 | 0.987 | 2.053 | 2.064 |
| donovan | NN-4 | 2.781 | 0.077 | -0.212 | 21.314 | 0.874 | 4.055 | 4.102 |
| shafter | NN-4 | 2.184 | 0.090 | 0.001 | 10.817 | 0.973 | 3.248 | 3.251 |
| elcajon | RF | 0.598 | 0.024 | 0.006 | 0.962 | 0.997 | 0.976 | 0.976 |
| donovan | RF | 1.341 | 0.037 | 0.014 | 4.938 | 0.971 | 1.988 | 1.988 |
| shafter | RF | 0.643 | 0.027 | 0.011 | 1.176 | 0.997 | 1.083 | 1.083 |

**Table C.3.** Level 0 train results for O3 (train and test on the same data set).

| Train Site | Test Site | Method | MAE | CvMAE | MBE | MSE | R^2 | crMSE | rMSE |
|---|---|---|---|---|---|---|---|---|---|
| elcajon | elcajon | MLR | 6.038 | 0.245 | 0.040 | 61.471 | 0.822 | 7.744 | 7.744 |
| donovan | donovan | MLR | 6.931 | 0.195 | -0.255 | 85.324 | 0.604 | 9.116 | 9.124 |
| shafter | shafter | MLR | 5.877 | 0.240 | 0.020 | 63.379 | 0.842 | 7.857 | 7.858 |
| elcajon | elcajon | NN-2 | 2.919 | 0.115 | -0.133 | 17.626 | 0.950 | 4.113 | 4.118 |
| donovan | donovan | NN-2 | 4.516 | 0.126 | -0.488 | 40.687 | 0.802 | 6.193 | 6.253 |
| shafter | shafter | NN-2 | 3.568 | 0.145 | 0.020 | 26.052 | 0.937 | 4.999 | 5.011 |
| elcajon | elcajon | NN-4 | 1.903 | 0.075 | -0.068 | 9.252 | 0.974 | 2.966 | 2.974 |
| donovan | donovan | NN-4 | 3.830 | 0.107 | -0.330 | 33.794 | 0.825 | 5.399 | 5.456 |
| shafter | shafter | NN-4 | 2.672 | 0.109 | -0.012 | 17.052 | 0.959 | 4.050 | 4.052 |
| elcajon | elcajon | RF | 1.217 | 0.049 | 0.015 | 3.987 | 0.988 | 1.987 | 1.987 |
| donovan | donovan | RF | 2.723 | 0.076 | -0.103 | 19.179 | 0.897 | 3.931 | 3.934 |
| shafter | shafter | RF | 1.284 | 0.054 | 0.024 | 4.651 | 0.988 | 2.155 | 2.155 |

**Table C.4.** Level 0 test results for O3.



| Train Site | Method | MAE | CvMAE | MBE | MSE | R^2 | crMSE | rMSE |
|---|---|---|---|---|---|---|---|---|
| elcajon | MLR | 3.869 | 0.256 | 0.015 | 28.999 | 0.683 | 5.333 | 5.333 |
| donovan | MLR | 5.573 | 0.556 | 0.160 | 75.037 | 0.315 | 8.504 | 8.512 |
| shafter | MLR | 4.671 | 0.353 | 0.004 | 38.467 | 0.511 | 6.191 | 6.191 |
| elcajon | NN-2 | 2.074 | 0.140 | 0.127 | 9.185 | 0.895 | 3.006 | 3.018 |
| donovan | NN-2 | 3.390 | 0.345 | 0.204 | 31.458 | 0.696 | 5.559 | 5.581 |
| shafter | NN-2 | 2.694 | 0.203 | 0.052 | 15.772 | 0.802 | 3.942 | 3.946 |
| elcajon | NN-4 | 1.465 | 0.098 | 0.076 | 5.524 | 0.938 | 2.327 | 2.331 |
| donovan | NN-4 | 2.739 | 0.288 | 0.105 | 24.875 | 0.729 | 4.907 | 4.924 |
| shafter | NN-4 | 2.089 | 0.158 | -0.060 | 10.702 | 0.865 | 3.252 | 3.256 |
| elcajon | RF | 0.972 | 0.064 | -0.028 | 2.929 | 0.968 | 1.683 | 1.683 |
| donovan | RF | 2.010 | 0.216 | 0.031 | 15.113 | 0.830 | 3.794 | 3.797 |
| shafter | RF | 1.028 | 0.078 | -0.041 | 3.822 | 0.951 | 1.943 | 1.943 |

**Table C.5.** Level 1 train results for NO2 (train and test on the same data set).



| Train Site | Test Site | Method | MAE | CvMAE | MBE | MSE | R^2 | crMSE | rMSE |
|---|---|---|---|---|---|---|---|---|---|
| elcajon | donovan | MLR | 7.994 | 0.841 | -4.447 | 119.171 | -0.304 | 8.782 | 10.639 |
| elcajon | shafter | MLR | 7.495 | 0.565 | -1.816 | 103.831 | -0.303 | 8.871 | 9.990 |
| donovan | elcajon | MLR | 6.383 | 0.436 | 1.998 | 69.142 | 0.179 | 6.801 | 8.203 |
| donovan | shafter | MLR | 8.860 | 0.676 | 0.639 | 132.238 | -0.711 | 8.201 | 11.351 |
| shafter | elcajon | MLR | 7.472 | 0.504 | 1.940 | 115.789 | -0.303 | 9.532 | 10.366 |
| shafter | donovan | MLR | 8.553 | 0.856 | 0.904 | 143.748 | -0.309 | 10.080 | 11.542 |
| elcajon | donovan | NN-2 | 6.552 | 0.688 | -1.875 | 98.026 | -0.063 | 8.628 | 9.641 |
| elcajon | shafter | NN-2 | 5.367 | 0.405 | -0.491 | 52.894 | 0.334 | 7.077 | 7.189 |
| donovan | elcajon | NN-2 | 9.960 | 0.649 | 2.435 | 282.631 | -1.896 | 13.732 | 14.872 |
| donovan | shafter | NN-2 | 8.567 | 0.662 | 2.822 | 173.652 | -1.359 | 10.145 | 11.805 |
| shafter | elcajon | NN-2 | 9.623 | 0.642 | 3.077 | 269.781 | -2.158 | 13.186 | 14.291 |
| shafter | donovan | NN-2 | 9.446 | 0.918 | 2.953 | 250.758 | -1.049 | 11.432 | 13.326 |
| elcajon | donovan | NN-4 | 6.164 | 0.632 | -1.301 | 83.675 | 0.163 | 8.663 | 9.103 |
| elcajon | shafter | NN-4 | 5.771 | 0.436 | -0.298 | 58.188 | 0.266 | 7.473 | 7.601 |
| donovan | elcajon | NN-4 | 7.702 | 0.500 | -0.698 | 132.834 | -0.385 | 10.342 | 10.622 |
| donovan | shafter | NN-4 | 7.947 | 0.614 | 1.850 | 148.190 | -0.995 | 10.032 | 10.690 |
| shafter | elcajon | NN-4 | 8.609 | 0.563 | -0.022 | 156.109 | -0.689 | 11.210 | 11.768 |
| shafter | donovan | NN-4 | 8.358 | 0.827 | -0.362 | 176.864 | -0.769 | 12.024 | 12.658 |
| elcajon | donovan | RF | 5.813 | 0.598 | -1.477 | 72.216 | 0.291 | 8.005 | 8.414 |
| elcajon | shafter | RF | 5.560 | 0.420 | -0.604 | 50.668 | 0.364 | 6.887 | 7.065 |
| donovan | elcajon | RF | 5.904 | 0.384 | -1.458 | 61.572 | 0.346 | 7.186 | 7.597 |
| donovan | shafter | RF | 6.579 | 0.505 | -1.846 | 79.515 | -0.048 | 7.700 | 8.645 |
| shafter | elcajon | RF | 7.182 | 0.487 | -0.148 | 198.620 | -1.665 | 10.882 | 11.411 |
| shafter | donovan | RF | 7.220 | 0.725 | -1.549 | 153.445 | -0.642 | 11.112 | 11.635 |

**Table C.6.** Level 1 test results for NO2.



| Train Site | Method | MAE | CvMAE | MBE | MSE | R^2 | crMSE | rMSE |
|---|---|---|---|---|---|---|---|---|
| elcajon | MLR | 6.038 | 0.245 | 0.040 | 61.471 | 0.822 | 7.744 | 7.744 |
| donovan | MLR | 6.931 | 0.195 | -0.255 | 85.324 | 0.604 | 9.116 | 9.124 |
| shafter | MLR | 5.877 | 0.240 | 0.020 | 63.379 | 0.842 | 7.857 | 7.858 |
| elcajon | NN-2 | 2.919 | 0.115 | -0.133 | 17.626 | 0.950 | 4.113 | 4.118 |
| donovan | NN-2 | 4.516 | 0.126 | -0.488 | 40.687 | 0.802 | 6.193 | 6.253 |
| shafter | NN-2 | 3.568 | 0.145 | 0.020 | 26.052 | 0.937 | 4.999 | 5.011 |
| elcajon | NN-4 | 1.903 | 0.075 | -0.068 | 9.252 | 0.974 | 2.966 | 2.974 |
| donovan | NN-4 | 3.830 | 0.107 | -0.330 | 33.794 | 0.825 | 5.399 | 5.456 |
| shafter | NN-4 | 2.672 | 0.109 | -0.012 | 17.052 | 0.959 | 4.050 | 4.052 |
| elcajon | RF | 1.217 | 0.049 | 0.015 | 3.987 | 0.988 | 1.987 | 1.987 |
| donovan | RF | 2.723 | 0.076 | -0.103 | 19.179 | 0.897 | 3.931 | 3.934 |
| shafter | RF | 1.284 | 0.054 | 0.024 | 4.651 | 0.988 | 2.155 | 2.155 |

**Table C.7.** Level 1 train results for O3 (train and test on the same data set).





| Train Site | Test Site | Method | MAE | CvMAE | MBE | MSE | R^2 | crMSE | rMSE |
|---|---|---|---|---|---|---|---|---|---|
| elcajon | donovan | MLR | 11.383 | 0.319 | 8.605 | 227.071 | -0.176 | 10.340 | 14.065 |
| elcajon | shafter | MLR | 9.812 | 0.415 | 5.474 | 182.938 | 0.509 | 11.107 | 12.920 |
| donovan | elcajon | MLR | 8.384 | 0.327 | 0.173 | 118.977 | 0.671 | 10.163 | 10.488 |
| donovan | shafter | MLR | 13.931 | 0.614 | 4.485 | 304.950 | 0.096 | 11.566 | 16.753 |
| shafter | elcajon | MLR | 9.819 | 0.400 | -0.489 | 187.609 | 0.448 | 12.064 | 13.055 |
| shafter | donovan | MLR | 13.205 | 0.373 | 6.385 | 321.685 | -0.639 | 13.259 | 16.624 |
| elcajon | donovan | NN-2 | 10.910 | 0.305 | 4.597 | 231.519 | -0.184 | 13.095 | 14.476 |
| elcajon | shafter | NN-2 | 8.799 | 0.358 | 0.850 | 138.822 | 0.659 | 11.204 | 11.562 |
| donovan | elcajon | NN-2 | 11.993 | 0.492 | -5.201 | 300.587 | 0.088 | 14.488 | 15.944 |
| donovan | shafter | NN-2 | 12.644 | 0.547 | -4.902 | 276.752 | 0.186 | 14.168 | 15.888 |
| shafter | elcajon | NN-2 | 14.346 | 0.630 | -7.165 | 565.238 | -0.780 | 17.245 | 19.883 |
| shafter | donovan | NN-2 | 16.290 | 0.447 | 0.309 | 533.943 | -1.188 | 15.973 | 20.250 |
| elcajon | donovan | NN-4 | 11.144 | 0.311 | 5.321 | 233.599 | -0.182 | 13.149 | 14.506 |
| elcajon | shafter | NN-4 | 9.151 | 0.376 | 1.102 | 148.600 | 0.623 | 11.621 | 12.024 |
| donovan | elcajon | NN-4 | 12.290 | 0.506 | -5.953 | 294.927 | 0.099 | 14.143 | 16.031 |
| donovan | shafter | NN-4 | 17.186 | 0.773 | -10.780 | 597.851 | -0.945 | 17.224 | 21.627 |
| shafter | elcajon | NN-4 | 11.177 | 0.480 | -3.281 | 271.195 | 0.156 | 14.606 | 15.313 |
| shafter | donovan | NN-4 | 13.084 | 0.372 | 3.730 | 325.781 | -0.556 | 15.852 | 17.251 |
| elcajon | donovan | RF | 10.679 | 0.302 | 6.487 | 189.558 | 0.051 | 11.079 | 13.496 |
| elcajon | shafter | RF | 9.739 | 0.401 | 1.367 | 157.403 | 0.601 | 11.999 | 12.406 |
| donovan | elcajon | RF | 11.458 | 0.469 | -4.232 | 206.904 | 0.381 | 12.335 | 13.735 |
| donovan | shafter | RF | 14.236 | 0.608 | -4.891 | 300.792 | 0.165 | 14.834 | 17.082 |
| shafter | elcajon | RF | 8.610 | 0.364 | -1.488 | 120.284 | 0.640 | 10.315 | 10.841 |
| shafter | donovan | RF | 11.045 | 0.311 | 6.332 | 182.582 | 0.156 | 11.390 | 13.459 |

**Table C.8.** Level 1 test results for O3.





| Train Sites | Method | MAE | CvMAE | MBE | MSE | R^2 | crMSE | rMSE |
|---|---|---|---|---|---|---|---|---|
| {donovan, shafter} | MLR | 5.277 | 0.416 | 0.046 | 56.139 | 0.400 | 7.429 | 7.429 |
| {donovan, elcajon} | MLR | 4.480 | 0.326 | 0.042 | 44.649 | 0.585 | 6.559 | 6.559 |
| {elcajon, shafter} | MLR | 4.829 | 0.332 | 0.021 | 43.007 | 0.514 | 6.531 | 6.531 |
| {donovan, shafter} | NN-2 | 2.935 | 0.231 | 0.212 | 23.237 | 0.755 | 4.755 | 4.765 |
| {donovan, elcajon} | NN-2 | 2.629 | 0.192 | 0.178 | 19.636 | 0.820 | 4.327 | 4.338 |
| {elcajon, shafter} | NN-2 | 2.665 | 0.183 | -0.014 | 15.152 | 0.829 | 3.871 | 3.877 |
| {donovan, shafter} | NN-4 | 2.026 | 0.159 | 0.079 | 12.776 | 0.865 | 3.531 | 3.537 |
| {donovan, elcajon} | NN-4 | 1.763 | 0.129 | 0.077 | 10.645 | 0.901 | 3.192 | 3.195 |
| {elcajon, shafter} | NN-4 | 1.823 | 0.126 | 0.070 | 8.225 | 0.906 | 2.857 | 2.860 |
| {donovan, shafter} | RF | 1.284 | 0.101 | -0.044 | 7.900 | 0.918 | 2.741 | 2.743 |
| {donovan, elcajon} | RF | 1.154 | 0.084 | -0.030 | 6.068 | 0.945 | 2.370 | 2.371 |
| {elcajon, shafter} | RF | 1.102 | 0.076 | -0.039 | 4.081 | 0.954 | 2.017 | 2.017 |

**Table C.9.** Level 2 train results for NO2 (train and test on the same data set).

| Train Sites | Test Site | Method | MAE | CvMAE | MBE | MSE | R^2 | crMSE | rMSE |
|---|---|---|---|---|---|---|---|---|---|
| {donovan, shafter} | elcajon | MLR | 5.880 | 0.408 | 1.644 | 65.552 | 0.202 | 7.514 | 7.901 |
| {donovan, elcajon} | shafter | MLR | 7.243 | 0.547 | -2.411 | 90.821 | -0.153 | 8.164 | 9.373 |
| {elcajon, shafter} | donovan | MLR | 7.312 | 0.759 | -2.242 | 101.530 | -0.043 | 8.570 | 9.915 |
| {donovan, shafter} | elcajon | NN-2 | 7.881 | 0.513 | -1.165 | 243.984 | -1.583 | 11.727 | 12.331 |
| {donovan, elcajon} | shafter | NN-2 | 5.013 | 0.380 | 0.434 | 47.193 | 0.402 | 6.699 | 6.829 |
| {elcajon, shafter} | donovan | NN-2 | 5.786 | 0.592 | -0.804 | 77.869 | 0.242 | 8.167 | 8.693 |
| {donovan, shafter} | elcajon | NN-4 | 8.579 | 0.554 | -1.420 | 281.996 | -1.869 | 12.447 | 12.951 |
| {donovan, elcajon} | shafter | NN-4 | 5.864 | 0.445 | 0.069 | 62.582 | 0.203 | 7.721 | 7.883 |
| {elcajon, shafter} | donovan | NN-4 | 5.991 | 0.606 | -0.629 | 92.061 | 0.068 | 9.122 | 9.332 |
| {donovan, shafter} | elcajon | RF | 5.510 | 0.376 | -0.670 | 61.713 | 0.255 | 7.271 | 7.561 |
| {donovan, elcajon} | shafter | RF | 5.312 | 0.402 | -0.096 | 47.764 | 0.396 | 6.733 | 6.863 |
| {elcajon, shafter} | donovan | RF | 5.533 | 0.567 | -0.883 | 74.962 | 0.255 | 8.271 | 8.562 |

**Table C.10.** Level 2 test results for NO2.



| Train Sites | Method | MAE | CvMAE | MBE | MSE | R^2 | crMSE | rMSE |
|---|---|---|---|---|---|---|---|---|
| {donovan, shafter} | MLR | 6.396 | 0.249 | -0.034 | 73.967 | 0.785 | 8.493 | 8.494 |
| {donovan, elcajon} | MLR | 6.702 | 0.227 | 0.005 | 75.618 | 0.788 | 8.640 | 8.640 |
| {elcajon, shafter} | MLR | 6.312 | 0.271 | 0.006 | 71.620 | 0.811 | 8.385 | 8.385 |
| {donovan, shafter} | NN-2 | 3.857 | 0.150 | -0.050 | 30.588 | 0.911 | 5.487 | 5.493 |
| {donovan, elcajon} | NN-2 | 3.721 | 0.127 | 0.169 | 28.691 | 0.919 | 5.332 | 5.344 |
| {elcajon, shafter} | NN-2 | 3.508 | 0.150 | 0.082 | 25.892 | 0.934 | 5.041 | 5.048 |
| {donovan, shafter} | NN-4 | 2.447 | 0.096 | 0.046 | 14.251 | 0.959 | 3.763 | 3.765 |
| {donovan, elcajon} | NN-4 | 2.355 | 0.080 | 0.116 | 13.973 | 0.961 | 3.716 | 3.721 |
| {elcajon, shafter} | NN-4 | 2.210 | 0.094 | 0.104 | 11.863 | 0.969 | 3.408 | 3.412 |
| {donovan, shafter} | RF | 1.499 | 0.059 | 0.069 | 6.184 | 0.982 | 2.480 | 2.482 |
| {donovan, elcajon} | RF | 1.466 | 0.050 | 0.041 | 5.897 | 0.984 | 2.421 | 2.422 |
| {elcajon, shafter} | RF | 1.325 | 0.057 | 0.023 | 4.921 | 0.987 | 2.216 | 2.216 |

**Table C.11.** Level 2 train results for O3 (train and test on the same data set).

| Train Sites | Test Site | Method | MAE | CvMAE | MBE | MSE | R^2 | crMSE | rMSE |
|---|---|---|---|---|---|---|---|---|---|
| {donovan, shafter} | elcajon | MLR | 8.981 | 0.362 | -1.747 | 136.139 | 0.607 | 10.070 | 11.263 |
| {donovan, elcajon} | shafter | MLR | 10.436 | 0.447 | 6.596 | 195.691 | 0.452 | 10.844 | 13.384 |
| {elcajon, shafter} | donovan | MLR | 11.842 | 0.332 | 8.646 | 234.924 | -0.168 | 10.887 | 14.470 |
| {donovan, shafter} | elcajon | NN-2 | 8.585 | 0.353 | -0.863 | 142.215 | 0.581 | 10.743 | 11.402 |
| {donovan, elcajon} | shafter | NN-2 | 8.227 | 0.338 | -0.202 | 120.049 | 0.694 | 10.390 | 10.844 |
| {elcajon, shafter} | donovan | NN-2 | 9.896 | 0.278 | 5.069 | 180.978 | 0.103 | 11.353 | 12.892 |
| {donovan, shafter} | elcajon | NN-4 | 9.708 | 0.391 | -1.786 | 187.381 | 0.466 | 12.179 | 12.983 |
| {donovan, elcajon} | shafter | NN-4 | 9.019 | 0.374 | -0.536 | 139.776 | 0.638 | 11.293 | 11.721 |
| {elcajon, shafter} | donovan | NN-4 | 9.802 | 0.274 | 4.557 | 159.778 | 0.249 | 11.398 | 12.544 |
| {donovan, shafter} | elcajon | RF | 7.892 | 0.327 | -1.715 | 100.997 | 0.702 | 9.286 | 9.875 |
| {donovan, elcajon} | shafter | RF | 9.568 | 0.397 | 0.597 | 150.607 | 0.613 | 11.533 | 12.148 |
| {elcajon, shafter} | donovan | RF | 9.133 | 0.259 | 4.811 | 135.414 | 0.351 | 9.986 | 11.571 |

**Table C.12.** Level 2 test results for O3.



| Train Sites | Method | MAE | CvMAE | MBE | MSE | R^2 | crMSE | rMSE |
|---|---|---|---|---|---|---|---|---|
| {donovan, elcajon, shafter} | MLR | 5.505 | 0.478 | -0.585 | 66.057 | 0.257 | 7.474 | 7.805 |
| {donovan, elcajon, shafter} | NN-2 | 3.205 | 0.276 | 0.024 | 26.908 | 0.711 | 4.929 | 4.967 |
| {donovan, elcajon, shafter} | NN-4 | 1.916 | 0.164 | 0.006 | 10.234 | 0.883 | 3.083 | 3.095 |
| {donovan, elcajon, shafter} | RF | 0.971 | 0.090 | -0.102 | 3.344 | 0.961 | 1.628 | 1.648 |

**Table C.13.** Level 3 train results for NO2 (train and test on the same data set).

| Train Sites | Test Site | Method | MAE | CvMAE | MBE | MSE | R^2 | crMSE | rMSE |
|---|---|---|---|---|---|---|---|---|---|
| {donovan, elcajon, shafter} | elcajon | MLR | 4.458 | 0.296 | 0.560 | 37.307 | 0.585 | 6.001 | 6.060 |
| {donovan, elcajon, shafter} | donovan | MLR | 6.819 | 0.707 | -2.001 | 91.429 | 0.074 | 8.431 | 9.428 |
| {donovan, elcajon, shafter} | shafter | MLR | 5.156 | 0.390 | -0.060 | 48.853 | 0.381 | 6.928 | 6.961 |
| {donovan, elcajon, shafter} | elcajon | NN-2 | 2.595 | 0.175 | -0.063 | 13.555 | 0.845 | 3.655 | 3.669 |
| {donovan, elcajon, shafter} | donovan | NN-2 | 4.108 | 0.420 | 0.050 | 47.008 | 0.556 | 6.660 | 6.765 |
| {donovan, elcajon, shafter} | shafter | NN-2 | 3.064 | 0.231 | 0.120 | 20.686 | 0.742 | 4.473 | 4.486 |
| {donovan, elcajon, shafter} | elcajon | NN-4 | 1.837 | 0.123 | -0.041 | 7.837 | 0.912 | 2.772 | 2.782 |
| {donovan, elcajon, shafter} | donovan | NN-4 | 3.167 | 0.335 | -0.075 | 38.583 | 0.542 | 5.784 | 5.812 |
| {donovan, elcajon, shafter} | shafter | NN-4 | 2.108 | 0.159 | 0.016 | 10.459 | 0.868 | 3.220 | 3.225 |
| {donovan, elcajon, shafter} | elcajon | RF | 1.079 | 0.072 | -0.039 | 3.634 | 0.959 | 1.885 | 1.886 |
| {donovan, elcajon, shafter} | donovan | RF | 2.583 | 0.277 | -0.302 | 20.818 | 0.768 | 4.453 | 4.495 |
| {donovan, elcajon, shafter} | shafter | RF | 1.358 | 0.103 | -0.035 | 5.324 | 0.933 | 2.281 | 2.287 |

**Table C.14.** Level 3 test results for NO2.

| Train Sites | Method | MAE | CvMAE | MBE | MSE | R^2 | crMSE | rMSE |
|---|---|---|---|---|---|---|---|---|
| {donovan, elcajon, shafter} | MLR | 7.893 | 0.277 | 1.474 | 117.623 | 0.509 | 9.553 | 10.226 |
| {donovan, elcajon, shafter} | NN-2 | 4.547 | 0.157 | 0.426 | 43.025 | 0.834 | 6.216 | 6.309 |
| {donovan, elcajon, shafter} | NN-4 | 2.509 | 0.088 | 0.174 | 14.705 | 0.938 | 3.611 | 3.660 |
| {donovan, elcajon, shafter} | RF | 1.379 | 0.044 | 0.308 | 5.251 | 0.976 | 1.865 | 1.936 |

**Table C.15.** Level 3 train results for O3 (train and test on the same data set).



| Train Sites | Test Site | Method | MAE | CvMAE | MBE | MSE | R^2 | crMSE | rMSE |
|---|---|---|---|---|---|---|---|---|---|
| {donovan, elcajon, shafter} | elcajon | MLR | 6.859 | 0.278 | -0.920 | 81.607 | 0.764 | 8.628 | 8.865 |
| {donovan, elcajon, shafter} | donovan | MLR | 9.870 | 0.276 | 5.047 | 169.555 | 0.141 | 10.236 | 12.282 |
| {donovan, elcajon, shafter} | shafter | MLR | 6.727 | 0.275 | 0.400 | 82.576 | 0.796 | 8.831 | 8.891 |
| {donovan, elcajon, shafter} | elcajon | NN-2 | 3.732 | 0.148 | -0.018 | 28.075 | 0.920 | 5.208 | 5.224 |
| {donovan, elcajon, shafter} | donovan | NN-2 | 5.826 | 0.162 | 1.373 | 65.155 | 0.690 | 7.610 | 7.934 |
| {donovan, elcajon, shafter} | shafter | NN-2 | 4.210 | 0.168 | -0.039 | 36.454 | 0.914 | 5.843 | 5.855 |
| {donovan, elcajon, shafter} | elcajon | NN-4 | 2.375 | 0.095 | -0.069 | 13.066 | 0.963 | 3.552 | 3.572 |
| {donovan, elcajon, shafter} | donovan | NN-4 | 4.541 | 0.126 | 1.132 | 46.867 | 0.757 | 6.182 | 6.402 |
| {donovan, elcajon, shafter} | shafter | NN-4 | 2.669 | 0.109 | -0.106 | 15.932 | 0.961 | 3.937 | 3.945 |
| {donovan, elcajon, shafter} | elcajon | RF | 1.391 | 0.056 | 0.019 | 5.064 | 0.985 | 2.233 | 2.234 |
| {donovan, elcajon, shafter} | donovan | RF | 3.504 | 0.096 | 1.142 | 28.621 | 0.849 | 4.594 | 4.837 |
| {donovan, elcajon, shafter} | shafter | RF | 1.853 | 0.072 | 0.105 | 8.391 | 0.980 | 2.775 | 2.783 |

**Table C.16.** Level 3 test results for O3.




## Appendix D:  Raw Results for the Split Neural Network Models

The following tables are error results for the split-NN models of size 3 and size 9. The error measures are labeled as MAE for mean absolute error, CvMAE for coefficient of variation of the mean absolute error, MBE for mean bias error, MSE for mean standard error, $R^2$ is the coefficient of determination, crMSE for centered root mean square error, and rMSE for root mean squared error. The results are disaggregated by train and test sites, and averaged across the sensor packages. However, because these are split models, both the the global model and the board-specific models are trained on all the sites. However, the trained board was not placed at the test site during training.

| Train Site | Test Site | MAE | CvMAE | MBE | MSE | R^2 | crMSE | rMSE |
|---|---|---|---|---|---|---|---|---|
| elcajon | donovan | 5.838 | 0.601 | -0.775 | 90.006 | 0.054 | 8.787 | 9.123 |
| elcajon | shafter | 5.246 | 0.397 | 0.284 | 51.872 | 0.345 | 6.909 | 7.142 |
| donovan | elcajon | 7.177 | 0.484 | -0.676 | 118.001 | -0.311 | 9.916 | 10.295 |
| donovan | shafter | 6.515 | 0.497 | 0.941 | 87.773 | -0.143 | 8.652 | 9.130 |
| shafter | elcajon | 7.544 | 0.484 | 0.452 | 183.866 | -0.923 | 10.592 | 11.094 |
| shafter | donovan | 7.516 | 0.736 | -0.530 | 155.259 | -0.307 | 10.295 | 11.056 |

**Table D.1.** Test results for split-NN level 1, size 3 (NO2).

| Train Site | Test Site | MAE | CvMAE | MBE | MSE | R^2 | crMSE | rMSE |
|---|---|---|---|---|---|---|---|---|
| elcajon | donovan | 5.713 | 0.590 | -1.012 | 78.898 | 0.206 | 8.326 | 8.800 |
| elcajon | shafter | 5.011 | 0.379 | 0.007 | 48.441 | 0.390 | 6.726 | 6.896 |
| donovan | elcajon | 6.426 | 0.436 | 0.016 | 88.722 | -0.018 | 8.797 | 9.180 |
| donovan | shafter | 6.272 | 0.478 | -0.493 | 78.929 | -0.028 | 8.441 | 8.760 |
| shafter | elcajon | 6.333 | 0.410 | 0.961 | 77.864 | 0.168 | 7.881 | 8.569 |
| shafter | donovan | 6.924 | 0.681 | -1.288 | 110.268 | 0.039 | 9.319 | 10.083 |

**Table D.2.** Test results for split-NN level 1, size 9 (NO2).





| Train Site | Test Site | MAE | CvMAE | MBE | MSE | R^2 | crMSE | rMSE |
|---|---|---|---|---|---|---|---|---|
| elcajon | donovan | 10.278 | 0.287 | 4.704 | 188.996 | 0.060 | 11.801 | 13.266 |
| elcajon | shafter | 8.280 | 0.336 | 0.862 | 125.486 | 0.692 | 10.668 | 10.982 |
| donovan | elcajon | 10.706 | 0.420 | -3.206 | 225.276 | 0.355 | 13.079 | 14.170 |
| donovan | shafter | 11.369 | 0.486 | -3.829 | 230.534 | 0.351 | 13.619 | 14.783 |
| shafter | elcajon | 10.857 | 0.480 | -3.101 | 380.840 | -0.227 | 14.472 | 15.351 |
| shafter | donovan | 12.195 | 0.343 | 4.319 | 302.175 | -0.297 | 14.332 | 15.918 |

**Table D.3.** Test results for split-NN level 1, size 3 (O3).

| Train Site | Test Site | MAE | CvMAE | MBE | MSE | R^2 | crMSE | rMSE |
|---|---|---|---|---|---|---|---|---|
| elcajon | donovan | 10.414 | 0.291 | 6.012 | 187.446 | 0.057 | 10.945 | 13.250 |
| elcajon | shafter | 8.234 | 0.335 | 1.909 | 123.068 | 0.696 | 10.482 | 10.872 |
| donovan | elcajon | 10.244 | 0.394 | -2.838 | 192.420 | 0.459 | 12.167 | 13.186 |
| donovan | shafter | 9.980 | 0.416 | -0.463 | 177.129 | 0.534 | 12.147 | 13.237 |
| shafter | elcajon | 9.709 | 0.423 | -2.282 | 211.003 | 0.344 | 12.567 | 13.295 |
| shafter | donovan | 11.240 | 0.317 | 5.503 | 216.113 | -0.003 | 12.947 | 14.428 |

**Table D.4.** Test results for split-NN level 1, size 9 (O3).

| Train Sites | Test Site | MAE | CvMAE | MBE | MSE | R^2 | crMSE | rMSE |
|---|---|---|---|---|---|---|---|---|
| {donovan, shafter} | elcajon | 5.915 | 0.392 | -1.035 | 91.805 | -0.013 | 8.458 | 8.739 |
| {donovan, elcajon} | shafter | 4.884 | 0.370 | 0.576 | 46.812 | 0.406 | 6.558 | 6.793 |
| {elcajon, shafter} | donovan | 5.362 | 0.543 | -0.373 | 73.628 | 0.302 | 8.108 | 8.411 |

**Table D.5.** Test results for split-NN level 2, size 3 (NO2).

| Train Sites | Test Site | MAE | CvMAE | MBE | MSE | R^2 | crMSE | rMSE |
|---|---|---|---|---|---|---|---|---|
| {donovan, shafter} | elcajon | 4.923 | 0.337 | -0.648 | 48.985 | 0.424 | 6.500 | 6.795 |
| {donovan, elcajon} | shafter | 4.749 | 0.360 | 0.676 | 43.165 | 0.453 | 6.221 | 6.497 |
| {elcajon, shafter} | donovan | 5.301 | 0.538 | -0.330 | 69.482 | 0.352 | 7.881 | 8.198 |

**Table D.6.** Test results for split-NN level 2, size 9 (NO2).



| Train Sites | Test Site | MAE | CvMAE | MBE | MSE | R^2 | crMSE | rMSE |
|---|---|---|---|---|---|---|---|---|
| {donovan, shafter} | elcajon | 8.285 | 0.336 | -1.515 | 139.473 | 0.596 | 10.573 | 11.204 |
| {donovan, elcajon} | shafter | 8.079 | 0.331 | -0.189 | 115.897 | 0.708 | 10.153 | 10.577 |
| {elcajon, shafter} | donovan | 9.356 | 0.262 | 4.033 | 155.842 | 0.250 | 11.020 | 12.172 |

**Table D.7.** Test results for split-NN level 2, size 3 (O3).

| Train Sites | Test Site | MAE | CvMAE | MBE | MSE | R^2 | crMSE | rMSE |
|---|---|---|---|---|---|---|---|---|
| {donovan, shafter} | elcajon | 7.434 | 0.297 | -0.910 | 105.619 | 0.695 | 9.443 | 9.977 |
| {donovan, elcajon} | shafter | 7.819 | 0.320 | 0.372 | 110.537 | 0.723 | 9.774 | 10.314 |
| {elcajon, shafter} | donovan | 9.022 | 0.253 | 4.190 | 141.869 | 0.313 | 10.427 | 11.654 |

**Table D.8.** Test results for split-NN level 2, size 9 (O3).