# Peer review of "Evaluating and Improving the Reliability of Gas-Phase Sensor System Calibrations Across New Locations for Ambient Measurements and Personal Exposure Monitoring"

_Atmospheric Measurement Techniques, 2019_

## Referee Comment (RC1) · Anonymous Referee #1 · 13 Mar 2019

This manuscript describes the assessment of several approaches that could be used to improve both the performance and the transferability of low cost gas phase sensor system calibrations. This is a crucial step in the enabling of these technologies for use air pollution monitoring, and this work is a valuable contribution to the growing body of literature on this major remaining challenge for these technologies. Previous work has demonstrated that although successful calibrations can be derived for low cost sensors through co-location with reference grade instruments, these calibrations do not hold if the sensors are moved to a new location, or even at the same location under

significantly different chemical or meteorological conditions, and are prone to model over-fitting. The lack of a robust and transferrable calibration strategy is most likely due to variations in the multiple environmental parameters, both chemical and physical, that effect sensor signals. The authors of this work propose that by using the data from multiple low cost sensors systems co-located with reference instruments in different locations the resultant calibration will be more generalized. This approach has been suggested previously, however, to this reviewers knowledge this is the most extensive investigation of this approach for gas phase electrochemical sensors to date. The authors also propose a novel two-stage "split-NN" approach to address the challenge of sensor to sensor variability when creating a global calibration.

The analysis presented in this manuscript is thorough and well written, and although the generalized calibration models developed still maintain large sensor errors the methods do show promise. I therefore recommend publication after the following minor comments have been addressed.

Minor comments:

1) Sect. 2.3 pg 9 lines13-15: It would be useful to the reader to know how much data was removed during the preprocessing steps.

2) Sect. 2.5: The split-NN is a novel approach for correcting for sensor-to-sensor variability in sensor signal and response to target compound concentrations. If I am not mistaken however, the environmental variables such as temperature are only used in the second stage of the process. As individual sensors are known to have different responses to their target compound it is more than likely that they will also differ in their responses to interfering compounds and environmental factors (this has been shown previously e.g. Smith et al. 2017). Would the authors not therefore get an improved result if the environmental parameters were included in both stages of the split-NN procedure? The authors should provide further justification of the variables chosen for each step in the split-NN.

3) Fig. 6: Needs units on y-axis.

4) Fig. 7: Needs units on plot axes and the time averaging used for the data points needs to be stated in the fig. caption.

5) Sect. 3.1 pg 15 lines 7-8: The sentence "The increase in bias is more pronounced in the higher capacity models" does not seem to be strongly supported by the data presented in Fig. 7. This statement needs supporting quantitatively or removing.

6) Sect. 3.2: It would be interesting to see the performance improvements from each stage of the split-NN approach. The addition of error plots similar to Fig. 7 for a single sensor after both stages of the process would help visualize the power of the approach.

7) Fig. 9: Needs units on y-axis.

8) Discussion: The authors are open about the limited success of the transferable calibration approaches investigated. It would, however, be beneficial to the field if the authors were to expand further on possible reasons for this and potential ways to improve the methods moving forward.

References: Smith K. R., Edwards P. M., Evans M. J., Lee J. D., Shaw M. D., Squires F., Wilde S. and Lewis A. C.: Clustering approaches to improve the performance of low cost air pollution sensors. Faraday Discuss, 15, 1-15, 2017.
* * *

---

## Referee Comment (RC2) · Christine Braban (Referee) · 17 Mar 2019

General overview:

This paper is a well thought through experiment and has some exciting ideas about the building of sensor networks and data processing to improve or understand error and bias in the systems. It uses a coherent approach, and develops a new statistical method which is mostly well described and accessible to the atmospheric scientist reader. There are a few major areas for improvements which are suggested below.

[Figure]

Major comments: 1) Presentation of measured data: Measurement data: despite doing a great job in statically analysis of sensor data, this paper lacks a figure with the epa and low cost sensor measurement data, ideally 1 panel with initial error envelope and 1 with final error envelope. For example no2 sensor measurement (+MAE+95th percentile MAE) (-MAE-95th percentile MAE) (I.e. from Figure 9 level2) - the authors may have better suggestions but if the paper and sensor systems data are to be used by scientists, citizens and community groups this is the information which needs to be presented. 2) Use of reference station data Generally the authors seem to treat the data from the reference station as not existing outside of the experiment and repeatedly make comments about the results in this paper showing new information about pollutant variation at the test sites. This shows a lack of forethought and analysis of existing data. The chemical climate for no2 and ozone at the sites is measured by the reference station, presumably for some years. The information to calculate the variability therefore is already collected and reported somewhere (e.g. EPA or San Diego authority reports) . The authors should use existing knowledge to inform their data analysis and interpretation rather than present the data in a knowledge vacuum. For example the speculation about the chemical climatology (my phrase not used in manuscript) of Shafer shows a clear disregard for existing evidence outside their experiment which is really avoidable.

3) Terminology: The error terminology is not applied consistently through the paper. "Error" is used a lot without it being clear whether it is overall error, expanded uncertainty, a replacement for Mean Absolute Error. Terminology is only introduced on p14 in the results section and then it is only sparsely used after that with terms such as centred error and bias error being added in (p15 line 6) This lack of clarity in the terminology leads to more questions than are answered in the manuscript e.g. • How do the authors assess the total error with the set of benchmarks • When the difference in error is discussed (e.g. p14 line 11), which difference is being discussed? • In the discussion the authors do not specify which error they are referring to e.g. p16 line 6 onwards. • Difference in error is discussed in qualitative terms despite there being

quantitative data in the work. Significance of differences are not discussed.

Overall the authors should revise with a consistent terminology and perhaps a table glossary. Also only MAE is shown in the main paper (Figure 6), please could the authors add the equivalent plots in the same figure for each of the benchmark errors. It is hard to assimilate the many tables in the Appendix with the results. The tables potentially should be moved into the main manuscript.

4) Discussion of cost Interestingly I think there are 2 points from this paper: the new N-N splitting method offers improvements for incrementally built networks and the Simple linear algorithmic gives the best understanding of changes. The former involves large amounts of expensive model development and potentially lack of clarity as to why the model works to improve sensor data (to a non-scientist sensor user). The cost implications for what is now not low cost at all (enclosures, powered fans, telemetry, expert algorithm development and maintenance) seem to be not openly explored, rather a nebulous "small cost increment for new nodes" offered as a positive: Could the authors perhaps discuss whether there is unconcious bias in their cost increment assessment?

Minor comments and corrections:

Abstract Could do with quantitative analysis of results in abstract e.g. how much does N-N improve model?

Introduction

Well written and readable. P4 lines 20-35: the list of the results does not fit in the introduction

Methods Section 2.1 sampling sites: the authors describe the sampling sites "expected profile" in descriptive terms. Given that the sites are regulatory local environmental and emission metadata for the site is probably available in the EPA station records. No references for the regulatory station information or data is provided or links to EPA reports using site data.

[Figure]

P6 line 13: "over the air" what does that mean?

P6 line 34: the noise on the signals and SD are discussed cf the sensor data. It would be useful to have the statistics of the raw (level 0) data and the "cleaned" or level 1 data for each sensor deployment as per epa site format used in Appendix B- or in a table in the main paper as it is critical for understanding the data processing effects

P7 line 6 is the sensiron accurate to 0.05% rh or is it just the data resolution on readout. Please could the accuracy/precision be stated. Resolution is not really useful.

Section 2.2.1 and 2.3 : the passive electrochemical samplers are placed in an actively ventilated housing "for this study". If truly for only this study and the sensors would be deployed differently under a normal operation, how are the results relevant to different setups?

2.3 Data is stored in the cloud: are they available for the public? What is the archive for data (and data identifier)?

P9 line 9: was the data from the reference station provisional or final ratified, I.e. regulatory automatic network data is ratified on a cycle. What was the date of data provision and data capture statistics for those periods? It is not enough to just say they came from the EPA. Data should be referenced properly.

Results: p14 line 15 is the difference between level 2 error vs level 1 statistically significant? It looks quite small on the Figure.

P9: averaging of minute data: arithmetic mean, time weighted average? How are data gaps in a minute treated?

Data filtering: filtering for "the realm of reasonable values" probably needs explaining more completely. Please list the QA filter steps in the appendix. Just to note, short lived plumes do not give reasonable values when you are normally used to looking at average values, but they may be real and relevant. What does +5V represent for each parameter? Probably the filtering is fine, but from the paper I cannot tell that.

P10 lines 1-5: you can tell that that would be overlap from the reference site data. Not necessary to confirm it with sensor measurements.

P15: It is interesting so see the change in bias between level 1 and 2 and I feel it should warrant further discussion in the manuscript. How many extra levels would be need to achieve an acceptable bias?

Figure 7: this figure needs a more complete caption to describe the graphs

P16 line 10: which error metric?

Figure 8: no x-axis label

Figure 9: please match scales on the two NO2 graphs and the two O3 graphs for ease of comparison. Also putting zero or an integer at the origin would be good practice.

P20 line 16. The authors mention using the 5s data to get more information and improve data quality despite the response time of the system likely to be not 5s (not quantified in this paper) and no reference given to show that this would a likely significant improvement rather than addition of more noise. It would be useful if the authors expanded on why they are optimistic about this.

Fig A1-A3: Figure captions could be more explanatory. What is the line vs the bars?

Appendix B too many decimal places in the tables!

---

## Author Comment (AC1) · 18 May 2019

Comment: "This manuscript describes the assessment of several approaches that could be used to improve both the performance and the transferability of low cost gas phase sensor system calibrations. This is a crucial step in the enabling of these technologies for use air pollution monitoring, and this work is a valuable contribution to the growing body of literature on this major remaining challenge for these technologies. Previous work has demonstrated that although successful calibrations can be derived for low cost sensors through co-location with reference grade instruments, these cali-

brations do not hold if the sensors are moved to a new location, or even at the same location under significantly different chemical or meteorological conditions, and are prone to model over-fitting. The lack of a robust and transferrable calibration strategy is most likely due to variations in the multiple environmental parameters, both chemical and physical, that effect sensor signals. The authors of this work propose that by using the data from multiple low cost sensors systems co-located with reference instruments in different locations the resultant calibration will be more generalized. This approach has been suggested previously, however, to this reviewers knowledge this is the most extensive investigation of this approach for gas phase electrochemical sensors to date. The authors also propose a novel two-stage "split-NN" approach to address the challenge of sensor to sensor variability when creating a global calibration. The analysis presented in this manuscript is thorough and well written, and although the generalized calibration models developed still maintain large sensor errors the methods do show promise. I therefore recommend publication after the following minor comments have been addressed."

Response: We thank reviewer 1 for their thoughtful and detailed comments. We believe we have completely addressed the reviewer's comments through revisions, as discussed below. We are grateful to reviewer 1 for their help in markedly improving the paper.

Minor Comments

Comment: "Sect. 2.3 pg 9 lines 13-15: It would be useful to the reader to know how much data was removed during the preprocessing steps."

Response: Thanks for noting this omission. We have added details about how much data was filtered to Section 2.4 Preprocessing, in particular, that 2.4% of the 5-second data was filtered.

Comment: "Sect. 2.5: The split-NN is a novel approach for correcting for sensor-to-sensor variability in sensor signal and response to target compound concentrations. If

I am not mistaken however, the environmental variables such as temperature are only used in the second stage of the process. As individual sensors are known to have different responses to their target compound it is more than likely that they will also differ in their responses to interfering compounds and environmental factors (this has been shown previously e.g. Smith et al. 2017). Would the authors not therefore get an improved result if the environmental parameters were included in both stages of the split-NN procedure? The authors should provide further justification of the variables chosen for each step in the split-NN."

Response: The environmental parameters contribute to training both stages through the process of backpropagation, as the sensor-specific model and the generic model are trained together. The concept is that, during training, the model-in-development generates a prediction, which determines an error with respect to the ground truth. Backpropagation pushes this error back into the network and the weights in the neurons in each layer are adjusted. As a consequence, even though the environmental parameters are injected downstream, their effects are felt upstream. We hesitated to provide this kind of detail in the article, leaving that for the cited literature on machine learning. However, if a more detailed account is desired, we'll be happy to accommodate.

Comment: "Fig. 6: Needs units on y-axis."

Response: Units have been added to the figure and the units for all metrics have been clarified in the text both where the metrics are introduced and in Appendix C.

Comment: "Fig. 7: Needs units on plot axes and the time averaging used for the data points needs to be stated in the fig. Caption."

Response: Units have been added to the figure. The details on minute-averaging have been elaborated in Section 2.4 Preprocessing, and apply to all the analyses, so we feel it is better centralized here. However, we believe that your comment was also directed at what each point in the target plot represents, so we have clarified that in the caption,

saying that each point in the plot corresponds to a different individual benchmark (i.e., a unique round, location, and board).

Comment: "Sect. 3.1 pg 15 lines 7-8: The sentence "The increase in bias is more pronounced in the higher capacity models" does not seem to be strongly supported by the data presented in Fig. 7. This statement needs supporting quantitatively or removing."

Response: True, nor do the results of the paper depend on this sentence. It has been removed.

Comment: "Sect. 3.2: It would be interesting to see the performance improvements from each stage of the split-NN approach. The addition of error plots similar to Fig. 7 for a single sensor after both stages of the process would help visualize the power of the approach."

Response: Unfortunately, a Split-NN provides ppb predictions only in the final stage. The earlier stage provides a set of latent variables that are learned from the input variables.

Comment: "Fig. 9: Needs units on y-axis."

Response: Units have been added to the figure.

Comment: "Discussion: The authors are open about the limited success of the transferable calibration approaches investigated. It would, however, be beneficial to the field if the authors were to expand further on possible reasons for this and potential ways to improve the methods moving forward."

Response: In the Conclusion we mention one direction for future work, using the higher resolution data of our sensor. We have now added two others (improvements to split-NN and use of infrastructure data). In the Discussion section, we do discuss some of these issues, including taking a closer look at bias error (3rd paragraph). But as of now, we're thinking of this as a pretty strong tradeoff between transferability and accuracy that can only be addressed through more diverse measurement (4th paragraph of Discussion). We still hold out some hope for split-NN as an economical approach to gaining more diverse measurement (5th paragraph).

References: Smith K. R., Edwards P. M., Evans M. J., Lee J. D., Shaw M. D., Squires F., Wilde S. and Lewis A. C.: Clustering approaches to improve the performance of low cost air pollution sensors. Faraday Discuss, 15, 1-15, 2017.

Response: Thank you. This work and several others were added in a related work paragraph in Section 2.6. Although the methods were trained similarly, it is important to note that in the intended use case, the MetaSense sensors would be generating a prediction about their current location, which would be wherever the end user happened to carry the sensor. It would be unlikely that they would be near other sensors.

---

## Author Comment (AC2) · 18 May 2019

Comment: "General overview: This paper is a well thought through experiment and has some exciting ideas about the building of sensor networks and data processing to improve or understand error and bias in the systems. It uses a coherent approach, and develops a new statistical method which is mostly well described and accessible to the atmospheric scientist reader. There are a few major areas for improvements which are suggested below."

[Figure]

Response: We thank reviewer 2 for their thoughtful and detailed review and suggestions. We have made substantial and detailed revisions to address the comments, as discussed below. We are grateful to reviewer 2 for their help in substantially improving the paper.

Major Comments

Comment: "Presentation of measured data: Measurement data: despite doing a great job in statically analysis of sensor data, this paper lacks a figure with the epa and low cost sensor measurement data, ideally 1 panel with initial error envelope and 1 with final error envelope. For example no2 sensor measurement (+MAE+95th percentile MAE) (-MAE-95th percentile MAE) (I.e. from Figure 9 level2) - the authors may have better suggestions but if the paper and sensor systems data are to be used by scientists, citizens and community groups this is the information which needs to be presented."

Response: Thanks for this idea. We have developed the suggested graphic and integrated it into Section 3.2 before the old Figure 9 (now Figure 10). To implement the before/after comparison, we used the collected voltages for the "before" data. If you have other or different ideas for the presentation and discussion, we are open to it. The new figure has also been attached below (Fig. 1).

Comment: "Use of reference station data Generally the authors seem to treat the data from the reference station as not existing outside of the experiment and repeatedly make comments about the results in this paper showing new information about pollutant variation at the test sites. This shows a lack of forethought and analysis of existing data. The chemical climate for no2 and ozone at the sites is measured by the reference station, presumably for some years. The information to calculate the variability therefore is already collected and reported somewhere (e.g. EPA or San Diego authority reports) . The authors should use existing knowledge to inform their data analysis and interpretation rather than present the data in a knowledge vacuum. For example

the speculation about the chemical climatology (my phrase not used in manuscript) of Shafer shows a clear disregard for existing evidence outside their experiment which is really avoidable."

Response: The authors thank the reviewer for this observation and have revised the manuscript to address this oversight. In Section 2.1, we have added information on the classification of sites as well as expected influences as defined by the respective air pollution control districts. Additionally, the authors have reviewed several documents from the respective air pollution control districts (including recent monitoring plans) in order to better understand the historic and typical pollutant levels and trends at these sites. A discussion of some of this information has also been added to Section 2.1, providing better context for our understanding of the data throughout the paper.

Comment: "Terminology: The error terminology is not applied consistently through the paper. "Error" is used a lot without it being clear whether it is overall error, expanded uncertainty, a replacement for Mean Absolute Error. Terminology is only introduced on p14 in the results section and then it is only sparsely used after that with terms such as centred error and bias error being added in (p15 line 6) This lack of clarity in the terminology leads to more questions than are answered in the manuscript e.g. âËŸA 'c How do the authors assess the total error with the set of benchmarks âËŸA 'c When the difference in error is discussed (e.g. p14 line 11), which difference is being discussed? âËŸA'c In the discussion the authors do not specify which error they are referring to e.g. p16 line 6 onwards. Difference in error is discussed in qualitative terms despite there being quantitative data in the work. Significance of differences are not discussed. Overall the authors should revise with a consistent terminology and perhaps a table glossary. Also only MAE is shown in the main paper (Figure 6), please could the authors add the equivalent plots in the same figure for each of the benchmark errors. It is hard to assimilate the many tables in the Appendix with the results. The tables potentially should be moved into the main manuscript."

Response: We apologize for causing confusion. We have made several edits throughout the paper to clarify which error we are referring to, when relevant. Although we present several errors in the tables to support direct comparison with multiple previous works, in general, we expect the various errors to track each other in relative, if not absolute, magnitude. Thus, any conclusion that we make using one error could be reached using one of the others.

Comment: "Discussion of cost Interestingly I think there are 2 points from this paper: the new NN splitting method offers improvements for incrementally built networks and the Simple linear algorithmic gives the best understanding of changes. The former involves large amounts of expensive model development and potentially lack of clarity as to why the model works to improve sensor data (to a non-scientist sensor user). The cost implications for what is now not low cost at all (enclosures, powered fans, telemetry, expert algorithm development and maintenance) seem to be not openly explored, rather a nebulous "small cost increment for new nodes" offered as a positive: Could the authors perhaps discuss whether there is unconcious bias in their cost increment assessment?"

Response: One of the goals of the low-cost sensing community is that eventually both the hardware and software will be available more or less "off the shelf", mitigating their costs for end-users. Today, however, many of the sensors and most of the accompanying software are research prototypes that are designed more for open-ended experimentation than end-use sensing. To support the ongoing transition to practice, we have now published an archival repository containing all of our hardware plans, software, and raw data, and cited it at the end of the Introduction. We have also made several changes to Section 2 to make it clearer what infrastructure is used for research versus calibration versus application. This includes an added picture on the right side of Figure 2 and an extended introduction to Section 2.3 Data Collection. If it is believed that the paper should be more explicit on the intended meaning of low-cost sensors, we could add a footnote to the Introduction.

Minor Comments and Corrections
Comment: "Abstract Could do with quantitative analysis of results in abstract e.g. how much does N-N improve model?"

Response: The results in the abstract have been made more precise both qualitatively and quantitatively, particularly for the split-NN model. In general, however, because so many models and training configurations were evaluated, a concise quantitative summary is difficult, and ultimately we found reference to the box plots served best. If there are ideas for a more concise quantitative summary, we are open to it.

Comment: "Introduction - Well written and readable. P4 lines 20-35: the list of the results does not fit in the introduction"

Response: We have moved the list of results to the conclusion (replacing and integrating the straight prose) and added additional detail supported by the body of the paper.

Comment: "Methods - Section 2.1 sampling sites: the authors describe the sampling sites "expected profile" in descriptive terms. Given that the sites are regulatory local environmental and emission metadata for the site is probably available in the EPA station records. No references for the regulatory station information or data is provided or links to EPA reports using site data."

Response: As described in response to the major comment regarding the use of reference station data, we have added information from the appropriate air pollution control districts, including references to official documents.

Comment: "P6 line 13: "over the air" what does that mean?"

Response: We apologize, it means wirelessly. We have replaced this phrase with something more descriptive.

Comment: "P6 line 34: the noise on the signals and SD are discussed cf the sensor data. It would be useful to have the statistics of the raw (level 0) data and the "cleaned" or level 1 data for each sensor deployment as per epa site format used in Appendix

B- or in a table in the main paper as it is critical for understanding the data processing effects"

Response: Thanks for noting this omission. We have added details about how much data was filtered to Section 2.4 Preprocessing, in particular that 2.4% of the 5-second data was filtered. It was not meaningful to report this in Appendix B, since that reports minute-level data.

Comment: "P7 line 6 is the sensiron accurate to 0.05% rh or is it just the data resolution on readout. Please could the accuracy/precision be stated. Resolution is not really useful."

Response: The accuracy of the humidity sensor was added. Thank you for the feedback.

Comment: "Section 2.2.1 and 2.3 : the passive electrochemical samplers are placed in an actively ventilated housing "for this study". If truly for only this study and the sensors would be deployed differently under a normal operation, how are the results relevant to different setups?"

Response: The expected use case for deployment for the sensors is that they will be exposed to ambient conditions and not placed in a larger enclosure with limited airflow. In real-world use cases, when the sensors are attached to backpacks, bikes, etc., the air flow would be sufficient for the sensors to sample ambient conditions. In our extended deployment, the sensors were placed inside of larger enclosures with small ports, so active ventilation was used to push air into the box. We have clarified these details in section 2.

Comment: "2.3 Data is stored in the cloud: are they available for the public? What is the archive for data (and data identifier)?"

Response: We have created a separate repository that is now linked in the paper in Section 6, after the acknowledgements. It also includes are hardware plans and

software.

Comment: "P9 line 9: was the data from the reference station provisional or final ratified, I.e. regulatory automatic network data is ratified on a cycle. What was the date of data provision and data capture statistics for those periods? It is not enough to just say they came from the EPA. Data should be referenced properly."

Response: The reference site data utilized in this analysis was not final ratified data as the timing of our study did not allow us to wait for this version of the data. Regarding the reference data, we did remove data collected during calibration periods as well as any data flagged during initial QA/QC by the regulatory agency who supplied the data. We have added these details to the end of Section 2.3 Data Collection. We have also added a note clarifying that the reference data used had not undergone complete QA/QC procedures and therefore is not final data from these stations in Section 6 Code and Data Availability.

Comment: "Results: p14 line 15 is the difference between level 2 error vs level 1 statistically significant? It looks quite small on the Figure.

Response: In this and the following sentence, we are emphasizing the *slight* improvement for Level 2, conveying that we feel that the effect size is small. Although the difference is likely significant given the size of the datasets, it would convey the wrong message since the effect is not especially impressive. However, we'd be glad to add this detail if it's desired.

Comment: "P9: averaging of minute data: arithmetic mean, time weighted average? How are data gaps in a minute treated?"

Response: To make the sentence in 2.4 Preprocessing clear on these questions, we rewrote it as follows: "For the remaining data, a simple average was computed over each one-minute window so as to match the time resolution of the data from the reference monitors. If an entire minute of data is missing due to a crashed sensor or
preprocessing, no minute-averaged value is generated."

Response: "Data filtering: filtering for "the realm of reasonable values" probably needs explaining more completely. Please list the QA filter steps in the appendix. Just to note, short lived plumes do not give reasonable values when you are normally used to looking at average values, but they may be real and relevant. What does +5V represent for each parameter? Probably the filtering is fine, but from the paper I cannot tell that."

Response: We believe we have sufficiently described these steps in section 2.4 Preprocessing, as they are simple threshold filters. The filtered data are not just spikes, but values that are simply not possible, either physically not possible or literally out of range for the sensor and represent a hardware or software failure.

Comment: "P10 lines 1-5: you can tell that that would be overlap from the reference site data. Not necessary to confirm it with sensor measurements.

Response: We apologize for the confusion generated by this section. Here we intended to state that the hypothesis (i.e., that the pollutant trends would vary between the different reference sites) had been verified by examining the distributions of the reference data. Furthermore, that the expected trends seemed to be reflected during the period of our deployment. The wording in Section 2.4 has been adjusted to clarify this point.

Comment: "P15: It is interesting so see the change in bias between level 1 and 2 and I feel it should warrant further discussion in the manuscript. How many extra levels would be need to achieve an acceptable bias?"

Response: This is a great observation by the reviewer, one that we feel emphasizes an important take-away in the paper. Based on the reviewer's comment we have enhanced the discussion in Section 3.1, following Figure 7 to better highlight the importance of error due to bias vs overall error. Regarding the question of how many levels would be necessary to achieve an acceptable bias, we feel that determining the precise number

of levels is somewhat beyond the scope of this paper - given that we do not have enough reference sites to continue exploring the question beyond 2 levels. That being said, this would be a valuable question to explore in future work.

Comment: "Figure 7: this figure needs a more complete caption to describe the graphs"

Response: A more detailed description of the target plots has been added to Figure 7.

Comment: "P16 line 10: which error metric?"

Response: We added "in MAE" to clarify that error is reported in MAE.

Comment: "Figure 8: no x-axis label"

Response: Fixed.

Comment: "Figure 9: please match scales on the two NO2 graphs and the two O3 graphs for ease of comparison. Also putting zero or an integer at the origin would be good practice."

Response: Thanks for catching this mistake. We have normalized the axes and 0-based them.

Comment: "P20 line 16. The authors mention using the 5s data to get more information and improve data quality despite the response time of the system likely to be not 5s (not quantified in this paper) and no reference given to show that this would a likely significant improvement rather than addition of more noise. It would be useful if the authors expanded on why they are optimistic about this."

Response: We didn't mean to express optimism, only potential. We have added a comment about the possible impacts of noise and the system response time, citing back to section 2.1.1.

Comment: "Fig A1-A3: Figure captions could be more explanatory. What is the line vs the bars?

Response: These have been clarified in the text. Each bar represents the total proportion of measurements at the given temperature or humidity (a histogram plot). The lines are a visualization of the kernel density estimation of the raw measurements.

Comment: "Appendix B too many decimal places in the tables!"

Response: Good point. We have trimmed them down to three decimal places to match the later tables and removed the decimals for the integer values.
* * *
**Prior to Model**  **After Model**

**Fig. 1.** A single board comparison of the relationship between the raw sensor values and target pollutant concentration.

[Figure]